# ConfPO: Exploiting Policy Model Confidence for Critical Token Selection in Preference Optimization

Hee Suk Yoon [1]  Eunseop Yoon [1]  Mark Hasegawa-Johnson [2]  Sungwoong Kim [3]  Chang D. Yoo [1]

## Abstract

We introduce ConfPO, a method for preference learning in Large Language Models (LLMs) that identifies and optimizes preference-critical tokens based solely on the training policy's confidence, without requiring any auxiliary models or compute. Unlike prior Direct Alignment Algorithms (DAAs) such as Direct Preference Optimization (DPO), which uniformly adjust all token probabilities regardless of their relevance to preference, ConfPO focuses optimization on the most impactful tokens. This targeted approach improves alignment quality while mitigating overoptimization (i.e., reward hacking) by using the KL divergence budget more efficiently. In contrast to recent token-level methods that rely on credit-assignment models or AI annotators, raising concerns about scalability and reliability, ConfPO is simple, lightweight, and model-free. Experimental results on challenging alignment benchmarks, including AlpacaEval 2 and Arena-Hard, demonstrate that ConfPO consistently outperforms uniform DAAs across various LLMs, delivering better alignment with zero additional computational overhead. The code is publicly accessible at https://github.com/hee-suk-yoon/ConfPO.

## 1. Introduction

Aligning Large Language Models (LLMs) with human preferences is commonly achieved through Reinforcement Learning from Human Feedback (RLHF) (Christiano et al., 2017; Ouyang et al., 2022; Yoon et al., 2024a), which fine-tunes a policy using a learned reward model. While effective, this approach introduces substantial computational overhead and is sensitive to reward model misspecification.

[1]Korea Advanced Institute of Science and Technology (KAIST), Republic of Korea, [2]University of Illinois Urbana-Champaign (UIUC), USA, [3]Korea University, Republic of Korea. Correspondence to: Chang D. Yoo <cd_yoo@kaist.ac.kr>.

*Proceedings of the 42ⁿᵈ International Conference on Machine Learning*, Vancouver, Canada. PMLR 267, 2025. Copyright 2025 by the author(s).

**Prompt**

Arrange the numbers 5, 13, 99, 1, and 22 in descending order. What is the first number in the new arrangement?

**Chosen**

The numbers arranged in descending order would be 99, 22, 13, 5, and 1. The first number in the new arrangement is 99.

**Rejected**

The numbers arranged in descending order would be: 22, 13, 99, 5, 1. So, the first number in the new arrangement is 1.

*Figure 1.* Highlighted tokens—green for the chosen, red for the rejected—are those regarded low-confidence by the policy model, meaning their predicted probability falls below the sequence average. This strategy surfaces tokens that carry high informational value or mark key decision points, while skipping predictable continuations. For example, in "22" (tokenized as '2', '2'), the first digit may be highlighted as uncertain, while the second, being a likely follow-up, is not. The incorrect answer "1" in the rejected response is another such low-confidence token, reflecting both its unpredictability and its role in distinguishing preference.

Direct Alignment Algorithms (DAAs) (Rafailov et al., 2024b; Azar et al., 2024; Xu et al., 2024; Ethayarajh et al., 2024; Hong et al., 2024; Park et al., 2024; Meng et al., 2024) offer a more efficient alternative by using the policy's own log-probabilities as an implicit reward signal. Among these, Direct Preference Optimization (DPO) (Rafailov et al., 2024b) has gained particular attention for optimizing the log-probability gap between chosen and rejected responses. SimPO (Meng et al., 2024), a recent variant of DPO, improves this framework by removing the need for a reference model and mitigating length bias, resulting in stronger alignment with human preferences. Yet, these methods assume all tokens contribute equally to preference, applying uniform optimization across the entire sequence without considering the varying informativeness of individual tokens.

A growing body of linguistic and cognitive research shows that words are not equally informative—some contribute little beyond confirming expectations, whereas others compel readers to revise their interpretation. Psycholinguistic *surprisal theory* (Hale, 2001; Jaeger & Levy, 2006; Smith & Levy, 2013) formalizes this insight by linking a word's informational value to its contextual predictability: the lower the conditional probability $P(w_i|\text{context})$, the higher the

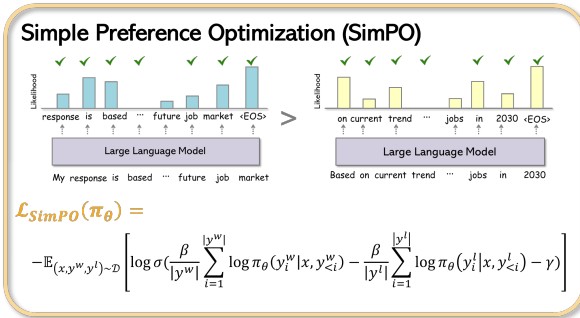
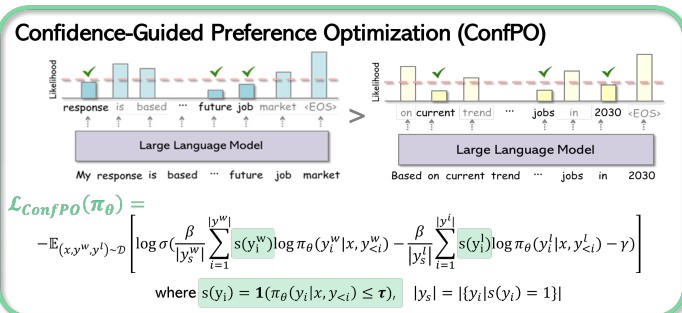

Figure 2. **Overview of the Confidence-guided Preference Optimization (ConfPO).** Existing DAA algorithm (SimPO (Meng et al., 2024)) uses all the tokens uniformly for preference alignment. ConfPO selectively optimizes on few tokens based on the policy model's confidence with no additional compute overhead.

surprisal and the greater the information conveyed. Modern LLMs embody this principle by explicitly estimating $P(w_i|context)$ for every token. Highly predictable tokens (high probability) add little new information, whereas low-probability tokens are information-rich and often determine how the rest of the sentence is interpreted (Merkx & Frank, 2021; Slaats et al., 2024). In essence, a small subset of low-confidence (high-surprisal) tokens carries most of the sentence-level information flow. We illustrate this idea in Figure 1, which shows that tokens with below-average conditional probability tend to be the unexpected, information-bearing tokens, whereas the remaining tokens are generally highly predictable continuations.

Motivated by these insights, we ask whether preference-relevant learning signals are better captured by selectively optimizing those low-confidence tokens. We provide empirical and theoretical analyses of gradient behavior in Direct Alignment Algorithms and show that a policy model's own confidence scores form a simple yet reliable proxy for identifying preference-critical tokens. Building on these findings, we introduce **ConfPO** (**Conf**idence-guided **P**reference **O**ptimization), a targeted extension of SimPO (Meng et al., 2024) that focuses optimization on the most informative tokens without auxiliary models or extra inference passes.

As illustrated in Figure 2, ConfPO dynamically filters out low-impact tokens, focusing updates only on those that meaningfully drive alignment. This targeted optimization strategy not only enhances performance on benchmarks such as AlpacaEval 2 and Arena-Hard, but also mitigates overoptimization by utilizing the KL budget more efficiently.

**In summary, our key contributions are:**

- **Efficient Selective-Token Alignment.** We propose a method that uses the policy model's confidence scores to identify tokens most critical for preference learning. Guided by empirical and theoretical gradient analyses of DAAs, this approach enables selective optimization without additional computational overhead (Section 4).

- **Improved Alignment Performance.** Across multiple LLMs and alignment benchmarks, ConfPO outperforms DAAs that uniformly train on all tokens (Table 1).

- **Overoptimization Mitigation.** By discarding tokens that contribute little to alignment, ConfPO makes more efficient use of the KL budget, reducing overoptimization (reward hacking) effects (Figure 7).

## 2. Related Works

**Reinforcement Learning from Human Feedback (RLHF)** Ensuring that models are helpful, safe, and factually accurate is a central goal of modern AI research (Yoon et al., 2022; 2023; 2024b;c; Yoon et al.). In particular, RLHF has become a widely used approach for aligning language models with human preferences. Early methods in this area often relied on online reinforcement learning techniques, such as PPO with a learned reward model, to adjust model behavior without requiring human input at every step (Christiano et al., 2017; Ouyang et al., 2022). However, these approaches are highly sensitive to hyperparameters and can be inefficient due to the need for extensive trajectory sampling. To address these challenges, Direct Alignment Algorithms (DAAs) have been developed as an alternative (Rafailov et al., 2024b; Azar et al., 2024; Xu et al., 2024; Ethayarajh et al., 2024; Hong et al., 2024; Park et al., 2024; Meng et al., 2024). DAAs eliminate the need for multi-stage training making the alignment process more efficient.

**Token-level DAAs** Sequence-level RLHF assigns a reward signal to the entire sequence, treating all tokens as equally contributing. However, this approach makes it challenging to obtain fine-grained learning signals. To address this limitation, recent efforts have focused on token-level reward signals for RLHF, particularly in DAAs. FIGA (Guo et al.) utilizes an external language model to generate data, which is then used for imitation learning. TDPO (Zeng et al.) enhances alignment and diversity by leveraging forward

KL divergence, while SePO (Yang et al., 2024) improves weak-to-strong generalization by pre-selecting tokens for preference learning using a weak model. T-REG (Zhou et al., 2024), in contrast, does not rely on external models but instead obtains token-level rewards by multiple forward passes through prompt augmentation. Despite these advancements, existing methods often introduce additional computational complexity, either through external models, extra training, or synthetic data generation. *In this paper, we propose an approach that enables token-level DAA while maintaining the computational efficiency of existing DAAs.*

**Overoptimizaion in RLHF**   Overoptimization in reward models has been identified as a critical issue in RLHF. Gao et al. (2023) highlighted that reward models in RLHF can suffer from overoptimization, leading to suboptimal alignment. Later, similar concerns were observed in DAAs, where Rafailov et al. (2024a) demonstrated that overoptimization can occur even without explicitly training a reward model. To mitigate this issue, Liu et al. (2024) proposed using a supervised loss as a regularizer, suggesting that it can help alleviate overoptimization. Additionally, Anonymous (2025) introduced an approach based on importance sampling, demonstrating its potential to mitigate the overoptimization problem. *We show in this paper that our proposed ConfPO tends to mitigate overoptimization due to the efficient use of the KL budget.*

# 3. Background and Problem Formulation

In this section, we provide an overview of preference learning for aligning large language models (LLMs) with human feedback and Direct Alignment Algorithms (DAAs).

## 3.1. RLHF: Preference Learning for Language Models

Reinforcement learning from human feedback (RLHF) (Christiano et al., 2017; Ouyang et al., 2022; Yoon et al., 2024a) is a commonly adopted approach to aligning large language models (LLMs) with human preferences. By leveraging annotations from humans (or AI systems), RLHF encourages LLMs to produce outputs that more closely match user expectations. Formally, let $\mathcal{D} = \{x^{(i)}, y^{w,(i)}, y^{l,(i)}\}_{i=1}^N$ be a preference dataset where each entry contains a prompt $x^{(i)}$ alongside two candidate responses: a preferred response $y^{w,(i)}$ and an unpreferred response $y^{l,(i)}$. The core of RLHF lies in a reward model $r^*(x, y)$, which assigns a scalar indicating how well a response $y$ aligns with human preferences for a given prompt $x$. A widely used model for pairwise preferences is the Bradley-Terry (BT) framework (Bradley & Terry, 1952), which specifies the probability that $y^w$ is preferred over $y^l$ as:

$$p(y^w \succ y^l \mid x) = \sigma\big(r^*(x, y^w) - r^*(x, y^l)\big). \quad (1)$$

RLHF proceeds in two stages: training the reward model and then using it to guide a policy model. In the first stage, the reward model $\hat{r}(x, y)$ is obtained via maximum likelihood estimation on the preference data. In the second stage, $\hat{r}(x, y)$ provides a feedback signal that drives optimization of the language model's policy $\pi_\theta$. Let $\mathcal{D}_x$ represent the distribution of prompts contained within $\mathcal{D}$. The objective can be written as:

$$\max_{\pi_\theta} \mathbb{E}_{x \sim \mathcal{D}_x,\, y \sim \pi_\theta(\cdot|x)} \Big[\hat{r}(x, y) \ - \ \beta \log \tfrac{\pi_\theta(\cdot|x)}{\pi_{\text{ref}}(\cdot|x)}\Big], \quad (2)$$

where $\beta$ is a coefficient controlling the penalty for deviating from a reference policy $\pi_{\text{ref}}$.

## 3.2. Direct Alignment Algorithms

Direct Alignment Algorithms (DAAs) offer an alternative to traditional RLHF by removing the need for explicit reward model training. Instead, these methods directly optimize the policy model using preference data, leveraging implicit signals from the model's own probability distributions.

**Direct Preference Optimization (DPO)**   DPO (Rafailov et al., 2024b) reparamerterizes the reward function $r$ using a closed-form expression as:

$$r_{\text{DPO}}(x, y) = \beta \log \frac{\pi_\theta(y \mid x)}{\pi_{\text{ref}}(y \mid x)} + \beta \log Z(x)$$

$$= \beta \sum_{i=1}^{|y|} \log \frac{\pi_\theta(y_i \mid x, y_{<i})}{\pi_{\text{ref}}(y_i \mid x, y_{<i})} + \beta \log Z(x). \quad (3)$$

where $\pi_\theta$ is the policy model, $\pi_{\text{ref}}$ is the reference policy, typically the supervised fine-tuned (SFT) model, and $Z(x)$ is the partition function. This reward formulation is incorporated into the BT ranking objective (Eq. 1), allowing DPO to express the probability of preference data directly with the policy model, rather than a separate reward model, yielding the following objective:

$$\mathcal{L}_{\text{DPO}}(\pi_\theta; \pi_{\text{ref}}) = -\mathbb{E}_{(x, y^w, y^l) \sim \mathcal{D}} \Big[ \log \sigma \big(\beta \sum_{i=1}^{|y^w|} \log \frac{\pi_\theta(y_i^w|x, y_{<i}^w)}{\pi_{\text{ref}}(y_i^w|x, y_{<i}^w)}$$

$$- \beta \sum_{i=1}^{|y^l|} \log \frac{\pi_\theta(y_i^l|x, y_{<i}^l)}{\pi_{\text{ref}}(y_i^l|x, y_{<i}^l)}\big)\Big]. \quad (4)$$

**Simple Preference Optimization (SimPO)**   The reward function in Eq. 3 presents several limitations. First, it relies on a reference model $\pi_{\text{ref}}$ during training, which is not required during inference. This creates an inconsistency between the training objective and the generation strategy at inference time. Second, it employs the summed log probabilities of tokens as the reward, which introduces length bias—longer sequences tend to have lower log probabilities. As a result, the model may overestimate probabilities for longer sequences to ensure that $y^w$ is assigned a higher reward than $y^l$.

To mitigate these issues, SimPO (Meng et al., 2024) redefines the reward function as follows:

$$r_{\text{SimPO}}(x, y) = \frac{\beta}{|y|} \log \pi_\theta(y \mid x) + \beta \log Z(x)$$

$$= \frac{\beta}{|y|} \sum_{i=1}^{|y|} \log \pi_\theta(y_i \mid x, y_{<i}) + \beta \log Z(x). \quad (5)$$

Using this implicit reward function, the objective for SimPO becomes the following:

$$\mathcal{L}_{\text{SimPO}}(\pi_\theta) = -\mathbb{E}_{(x,y^w,y^l)\sim\mathcal{D}}\Big[ \log \sigma \big( \frac{\beta}{|y^w|} \sum_{i=1}^{|y^w|} \log \pi_\theta(y_i^w|x, y_{<i}^w)$$

$$- \frac{\beta}{|y^l|} \sum_{i=1}^{|y^l|} \log \pi_\theta(y_i^l|x, y_{<i}^l) - \gamma) \Big], \quad (6)$$

where $\gamma$ is a target margin introduced by Meng et al. (2024) to ensure that the reward for the preferred response $r_{\text{SimPO}}(x, y^w)$ exceeds the reward for the unpreferred response $r_{\text{SimPO}}(x, y^l)$ by at least $\gamma$. SimPO demonstrates significant improvements on various representative benchmarks (i.e., AlphacaEval 2 (Li et al., 2023) and Arena-Hard (Li* et al., 2024)) compared to DPO, whereas many other variants of DPO (Yuan et al., 2023; Zhao et al., 2023; Azar et al., 2024; Xu et al., 2024; Ethayarajh et al., 2024; Hong et al., 2024; Park et al., 2024) fail to show a consistent performance advantage over the standard DPO. ***Thus, our method is built upon SimPO; however, in Section 7.2, we also demonstrate that our findings generalize to DPO.***

### 3.3. Problem Formulation: Token Selection for Preference Learning

Although Direct Alignment Algorithms (DAAs), such as DPO (Eq. 4) and SimPO (Eq. 6), have demonstrated strong performance in aligning LLMs with human preferences, they are typically optimized over all tokens in the training dataset. However, prior studies (Lin et al., 2024; Chen et al., 2024; Lai et al., 2024; Yoon et al., 2024a) have shown that not all tokens contribute equally to preference alignment. Uniformly optimizing over all tokens can introduce noise into the training process.

**Selective Token Reward Formulation** To address these challenges, we propose improving DAAs by optimizing only on effective tokens for preference learning. Specifically, we aim to identify and leverage a subset of tokens that are critical to aligning with human preferences, without incurring additional computational costs compared to standard algorithms like SimPO (Meng et al., 2024).

We extend the implicit reward formulation of SimPO (Eq. 5) by incorporating a selective token scoring function. The reward for a response $y$ is redefined as the joint log probability of critical (i.e., selective) tokens, expressed as:

$$r(x, y) = \frac{\beta}{|y_s|} \sum_{i=1}^{|y|} s(y_i) \log \pi_\theta(y_i \mid x, y_{<i}) + \beta \log Z(x), \quad (7)$$

where $s(y_i)$ is a selection function that determines whether to include the token $y_i$ for reward calculation, and $|y_s|$ denotes the total number of selected tokens. Incorporating this selective token-based reward formulation, we refine the preference optimization objective of SimPO (Eq. 6) as follows:

$$\mathcal{L}(\pi_\theta) =$$

$$-\mathbb{E}_{(x,y^w,y^l)\sim\mathcal{D}}\Big[ \log \sigma \big( \frac{\beta}{|y_s^w|} \sum_{i=1}^{|y^w|} s(y_i^w) \log \pi_\theta(y_i^w|x, y_{<i}^w)$$

$$- \frac{\beta}{|y_s^l|} \sum_{i=1}^{|y^l|} s(y_i^l) \log \pi_\theta(y_i^l|x, y_{<i}^l) - \gamma) \Big]. \quad (8)$$

While previous works have proposed token selection strategies, these approaches often involve significant computational overhead. Examples include performing multiple forward passes with augmented prompts (Zhou et al., 2024), leveraging annotations from powerful LLMs such as GPT-4 (Lai et al., 2024; Yoon et al., 2024a), or training auxiliary models to guide token selection (Yang et al., 2024). These methods, while effective, are complex and expensive, limiting their practicality.

***Unlike prior approaches, we aim to derive $s(y_i)$ directly from the internal signals of the model during training, ensuring that the computational cost remains identical to that of the original DAAs.***

## 4. Revisiting the Gradients of Direct Alignment Algorithms (DAAs)

In this section, we introduce a series of observations revealing how to use the internal signals of the training policy model to select important tokens for preference learning. We focus on the SimPO (Meng et al., 2024) due to its state-of-the-art performance, but we show that these observations also hold for DPO (Rafailov et al., 2024b) in Section 7.2.

**Gradient of SimPO** As illustrated in Meng et al. (2024), the gradients of SimPO (Eq. 6) can be written as follows:

$$\nabla_\theta \mathcal{L}_{\text{SimPO}}(\pi_\theta) = -\beta \mathbb{E}_{(x,y^w,y^l)\sim\mathcal{D}}\bigg[ s_\theta \bigg( \frac{1}{|y^w|} \nabla_\theta \log \pi_\theta(y^w|x)$$

$$- \frac{1}{|y^l|} \nabla_\theta \log \pi_\theta(y^l|x) \bigg) \bigg], \quad (9)$$

where $\quad s_\theta = \sigma\big( \frac{\beta}{|y^l|} \log \pi_\theta(y^l|x) - \frac{\beta}{|y^w|} \log \pi_\theta(y^w|x) + \gamma \big).$

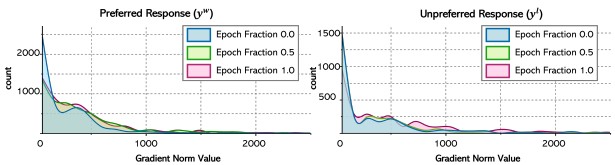

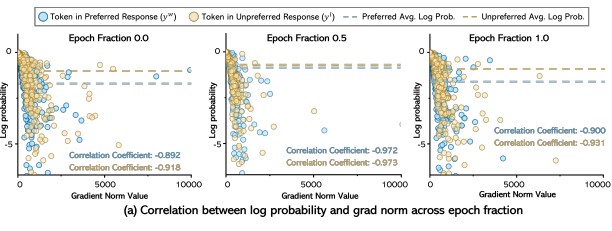

*Figure 3.* **Observation 1.** Token-level gradient norm values (i.e., $\|\nabla_\theta \log \pi_\theta(y_i|x, y_{<i})\|$) follow a long-tailed distribution for both the preferred response (*left*) and the unpreferred response (*right*). Each distribution is computed from 10 sampled sentences, and Llama-3-Base (8B) was used. This observation shows that not all tokens contribute equally to preference alignment.

In this gradient equation, the sequence-level gradients, $\nabla_\theta \log \pi_\theta(y^w|x)$ and $\nabla_\theta \log \pi_\theta(y^l|x)$ govern how the model adjusts probabilities across the tokens. To better understand the token-level contributions to the gradient, we can decompose these sequence-level terms (Eq. 10). For brevity, we focus on the gradient with respect to a single response $y$ (either $y^w$ or $y^l$), as the expressions for both are structurally identical.

$$\nabla_\theta \log \pi_\theta(y \mid x) = \sum_{i=1}^{|y|} \nabla_\theta \log \pi_\theta(y_i|x, y_{<i}). \quad (10)$$

### 4.1. Analyzing Token-Level Gradients in Direct Alignment Algorithms (DAAs)

**Observation 1: Token-Level Gradients Follow a Long-Tailed Distribution** To understand how individual tokens contribute to preference alignment, we analyze the gradient norm $\|\nabla_\theta \log \pi_\theta(y_i|x, y_{<i})\|$ for each token in both preferred ($y^w$) and unpreferred ($y^l$) responses over the course of training (see Figure 3). Our findings reveal a clear long-tailed distribution: although most tokens exhibit minimal gradient updates, a small subset accounts for disproportionately large gradients, indicating that these tokens drive the bulk of the learning signal. This observation aligns with earlier work on language model pretraining (Lin et al., 2024), which similarly reports that focusing on a fraction of critical tokens can significantly enhance model performance. *In the context of preference learning, the prevalence of low-gradient tokens may indicate their limited relevance to human preference signals, diluting the model's overall effectiveness when training indiscriminately on all tokens.*

**Observation 2: There is a High Correlation between the Gradient Norm and the Confidence of a Token** Building on Observation 1, one potential strategy to improve preference learning might be to prioritize tokens with large gradient norms. However, directly computing $\|\nabla_\theta \log \pi_\theta(y_i|x, y_{<i})\|$ for every token is prohibitively expensive, since it requires a dedicated backward pass per token—an approach that would greatly exceed the computational budget of standard preference learning.

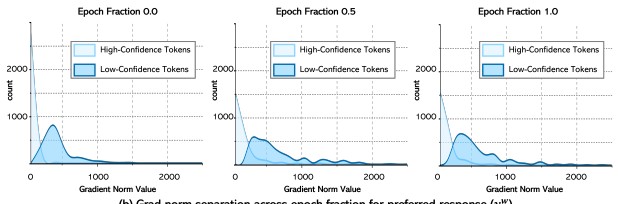

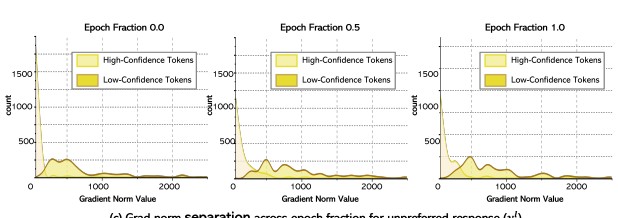

(a) Correlation between log probability and grad norm across epoch fraction

(b) Grad norm separation across epoch fraction for preferred response ($y^w$)

(c) Grad norm **separation** across epoch fraction for unpreferred response ($y^l$)

*Figure 4.* **Observation 2.** (a) Spearman correlation plot between the policy model's log probability (confidence) for each token, $\log \pi_\theta(y_i|x, y_{<i})$, and its gradient norm, $\|\nabla_\theta \log \pi_\theta(y_i|x, y_{<i})\|$, over the course of training. A strong negative correlation emerges for tokens in both the preferred ($y^w$) and unpreferred ($y^l$) response, indicating that tokens with lower probabilities tend to have larger gradients. (b-c) For both $y^w$ and $y^l$, the tokens are split into two groups based on whether their probabilities exceed or fall below the average probability. We then illustrate the gradient norm distributions for each group, revealing a distinct gap throughout training: high-confidence tokens generally show smaller gradients, while low-confidence tokens exhibit consistently larger gradients.

To address this, we look for a simpler proxy that can be measured without extra forward or backward passes. As shown in Figure 4-(a), we find a strong negative correlation between the model's probability (confidence) assigned to a token, $\pi_\theta(y_i|x, y_{<i})$, and the magnitude of its gradient norm, $\|\nabla_\theta \log \pi_\theta(y_i|x, y_{<i})\|$. To further analyze this relationship, we group tokens based on whether their probability exceeds or falls below the average probability. Figure 4(b-c) presents the gradient norm distributions for each group, revealing a clear distinction: high-confidence tokens consistently exhibit smaller gradients, while low-confidence tokens have significantly larger gradients, for both the preferred ($y^w$) and unpreferred ($y^l$) responses.

These findings suggest that *tokens with the high learning signals (i.e, those critical for preference alignment) can be effectively identified using only the model's confidence scores*. Importantly, this approach maintains the *exact same computational cost* as standard preference learning, making it both practical and scalable. *A theoretical account of this relationship is presented later in Section 4.2.*

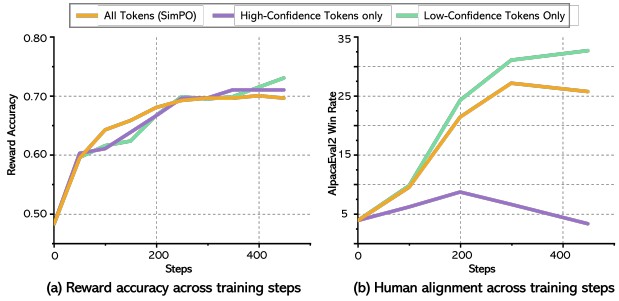

*Figure 5.* **Observation 3.** We conduct SimPO learning using three settings: (i) all tokens, (ii) only high-confidence tokens, and (iii) only low-confidence tokens. (a) Each setting calculates its reward based on the selected tokens, and all three eventually reach high accuracy—indicating that, within their respective token subsets, the model discriminates well between preferred and unpreferred responses. (b) Despite converging to a similar reward accuracy, training solely on high-confidence tokens yields no gain in alignment with human preferences. This suggests that high-confidence tokens carry limited preference information, whereas low-confidence tokens more strongly drive alignment improvements.

**Observation 3: Low-Confidence Tokens Dominate Preference Learning**   Building on Observation 2, we further explore whether low-confidence (high-gradient) tokens indeed provide more critical preference information than high-confidence (low-gradient) tokens. Figure 5 compares human alignment outcomes (via the AlpacaEval2 (Li et al., 2023) Win Rate) under three conditions: training on only high-confidence tokens, only low-confidence tokens, and all tokens. Despite all three approaches achieving high training reward accuracy within their respective token subsets, their human alignment trajectories differ significantly:

1. **High-Confidence Tokens Only:** This yields negligible or even *negative* improvements, suggesting that highly confident tokens offer limited learning signal and may introduce noise into the optimization.

2. **Low-Confidence Tokens Only:** Restricting training to these high-gradient tokens significantly boosts alignment, surpassing even the baseline that uses *all* tokens.

These findings align with earlier work (Lin et al., 2024) showing that **not all tokens are equally valuable**. In fact, high-confidence tokens can impede preference learning by overshadowing more informative signals from their low-confidence counterparts. Consequently, **selectively emphasizing low-confidence tokens** appears to be a straightforward yet powerful way to enhance preference alignment.

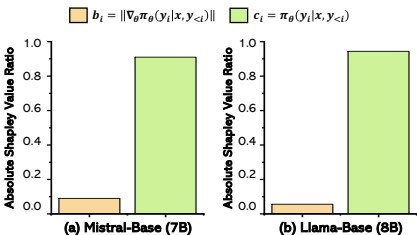

*Figure 6.* Absolute Shapley value decomposition of the gradient-norm ratio $r_i = \|\nabla_\theta \pi_\theta(y_i \mid x, y_{<i})\| / \pi_\theta(y_i \mid x, y_{<i})$, illustrating the relative contributions of its numerator ($b_i$) and denominator ($c_i$). Results are shown for both Mistral-Base (7B) and Llama-Base (8B), where $c_i$ consistently accounts for a larger share of the total attribution.

### 4.2. Theoretical Rationale for Why Low-Confidence Tokens Carry Strong Learning Signals

Looking back at the log-probability gradient norm of a single token $\nabla_\theta \log \pi_\theta(y_i|x, y_{<i})$, this can be equally expressed as follows using the chain rule:

$$\|\nabla_\theta \log \pi_\theta(y_i|x, y_{<i})\| = \frac{\|\nabla_\theta \pi_\theta(y_i|x, y_{<i})\|}{\pi_\theta(y_i|x, y_{<i})}. \quad (11)$$

Thus, if token confidence $\pi_\theta(y_i|x, y_{<i})$ dominates this ratio, the log-probability gradient norm $\|\nabla_\theta \log \pi_\theta(y_i|x, y_{<i})\|$ decreases as confidence increases, establishing the observed negative correlation.

**Shapley Value Analysis**   To formalize the dominant role of the token probability $\pi_\theta(y_i|x, y_{<i})$ in determining the log-probability gradient norm $\|\nabla_\theta \log \pi_\theta(y_i|x, y_{<i})\|$, we consider the ratio $r_i = \frac{b_i}{c_i}$, where $b_i = \|\nabla_\theta \pi_\theta(y_i|x, y_{<i})\|$ and $c_i = \pi_\theta(y_i|x, y_{<i})$. The objective is to quantify the extent to which each input, $b_i$ and $c_i$, contributes to variation in the output $r_i$.

To fairly quantify these contributions—particularly since $b_i$ and $c_i$ are not independent but are both functions of $\theta$—we use *Shapley values* (Shapley, 1953; Lundberg & Lee, 2017), a concept from cooperative game theory that offers a principled approach to attributing a model's output to its individual input features. While the original framework assumes feature independence, recent extensions account for dependencies between input features (Owen & Prieur, 2017; Aas et al., 2021). In particular, we adopt the methodology introduced by Aas et al. (2021)[1], which accurately estimates feature contributions under feature dependence. As shown in Figure 6, the results demonstrate that the denominator $c_i$ contributes substantially more to the variation in $r_i$ than the numerator $b_i$, providing theoretical support for the empirical observation that low-confidence tokens ($\pi_\theta(y_i|x, y_{<i})\downarrow$) tend to have stronger learning signals ($\|\nabla_\theta \log \pi_\theta(y_i|x, y_{<i})\|\uparrow$).

---

[1]https://github.com/NorskRegnesentral/shapr

# 5. ConfPO: Confidence-Guided Critical Token Selection for Preference Optimization

Motivated by our observations and theoretical rationale in Section 4, we introduce our proposed confidence-guided critical token selection for preference optimization (**ConfPO**).

**Confidence-guided Selective Token Reward Formulation** Using the fact that low confident tokens contain high learning information for preference learning, we propose the following implicit reward as a continuation for the selective token reward formulation that was introduced in Eq. 7:

$$r_{\text{ConfPO}}(x, y) = \frac{\beta}{|y_s|} \sum_{i=1}^{|y|} s(y_i) \log \pi_\theta(y_i \mid x, y_{<i}) + \beta \log Z(x),$$
(12)

where $s(y_i) = \mathbf{1}(\pi_\theta(y_i|x, y_{<i}) \leq \tau)$ and $|y_s| = |\{y_i|s(y_i) = 1\}|$.

Then, the resulting preference optimization objective of ConfPO can be expressed as:

$$\mathcal{L}_{\text{ConfPO}}(\pi_\theta) =$$

$$-\mathbb{E}_{(x,y^w,y^l)\sim\mathcal{D}} \Big[ \log \sigma \big( \frac{\beta}{|y_s^w|} \sum_{i=1}^{|y^w|} s(y_i^w) \log \pi_\theta(y_i^w|x, y_{<i}^w)$$

$$- \frac{\beta}{|y_s^l|} \sum_{i=1}^{|y^l|} s(y_i^l) \log \pi_\theta(y_i^l|x, y_{<i}^l) - \gamma \big) \Big],$$
(13)

where $s(y_i^w) = \mathbf{1}(\pi_\theta(y_i^w|x, y_{<i}^w) \leq \tau^w)$, $|y_s^w| = |\{y_i^w|s(y_i^w) = 1\}|$, $s(y_i^l) = \mathbf{1}(\pi_\theta(y_i^l|x, y_{<i}^l) \leq \tau^l)$, and $|y_s^l| = |\{y_i^l|s(y_i^l) = 1\}|$.

We define the confidence thresholds $\tau^w$ and $\tau^l$ as the average token probabilities across the respective responses:

$$\tau^w = \frac{1}{|y^w|} \sum_{i=1}^{|y^w|} \pi_\theta(y_i^w|x, y_{<i}^w), \quad \tau^l = \frac{1}{|y^l|} \sum_{i=1}^{|y^l|} \pi_\theta(y_i^l|x, y_{<i}^l).$$
(14)

In Appendix B, we empirically demonstrate that this thresholding strategy outperforms other thresholding choices.

# 6. Experimental Setup

## 6.1. Models and Datasets

Following Meng et al. (2024), we conduct experiments using two families of models: Mistral-7B (Jiang et al., 2023) and Llama-3-8B (AI@Meta, 2024), under both **Base** and **Instruct** configurations. For the **Base** setup, we utilize base models that have been fine-tuned via Supervised Fine-Tuning (SFT) on the UltraChat-200k dataset (Ding et al., 2023) (i.e., alignment-handbook/zephyr-7b-sft-full and

princeton-nlp/Llama-3-Base-8B-SFT). We then apply preference optimization using the UltraFeedback dataset (Cui et al., 2023). For the **Instruct** setup, we begin with instruction-tuned models (i.e., mistralai/Mistral-7B-Instruct-v0.2 and meta-llama/Meta-Llama-3-8B-Instruct) as the SFT models. Then, we use the preference data generated by Meng et al. (2024) which is derived from prompts in Ultrafeedback dataset and scored using llm-blender/PairRM reward model. *The full implementation details can be found in Appendix D.*

## 6.2. Evaluation Benchmarks

We assess models on two instruction-following benchmarks: **AlpacaEval 2** (Li et al., 2023) and **Arena-Hard v0.1** (Li* et al., 2024). AlpacaEval 2 consists of 805 questions from five datasets, while Arena-Hard v0.1 is an extension of MT-Bench (Zheng et al., 2023), comprising 500 problem-solving queries. For AlpacaEval 2, we report both the win rate (WR) and the length-controlled win rate (LC) (Dubois et al., 2024), where the LC is designed to reduce the length bias. For AlpacaEval 2, we use `alpaca_eval_gpt4_turbo_fn` annotator, which has a higher human agreement score then `weighted_alpaca_eval_gpt4_turbo` used in Meng et al. (2024).

## 6.3. Baselines

Similar to Meng et al. (2024), we compare ConfPO with other Direct Alignment Algorithms (DAAs) listed in Table 5 of the Appendix. **RRHF** (Yuan et al., 2023) and **SLiC-HF** (Zhao et al., 2023) use ranking-based losses, with RRHF normalizing log-likelihood by length, while SLiC-HF directly optimizes log-likelihood alongside an SFT objective. **IPO** (Azar et al., 2024) addresses the assumption of **DPO** (Rafailov et al., 2024b) that pairwise preferences can be treated as pointwise rewards. **CPO** (Xu et al., 2024) optimizes sequence likelihood while training with SFT. **KTO** (Ethayarajh et al., 2024) learns from non-paired preference data, and **ORPO** (Hong et al., 2024) removes the need for a reference model by introducing an odds ratio term to contrast winning and losing responses. **R-DPO** (Park et al., 2024) adds a regularization term to DPO to mitigate length bias. **SimPO** (Meng et al., 2024) resolves the discrepancy between the training objective and inference in DPO while also mitigating its length bias, resulting in a significant improvement in performance.

For the SimPO baseline, we retrained the model using the hyperparameters specified in their original paper (Meng et al., 2024). For other DAA baselines, we utilized the publicly available checkpoints provided by Meng et al. (2024). The hyperparameters used for ConfPO are listed in Table 6. Notably, we observe that *ConfPO generally favors a lower optimal $\beta$ and a higher $\gamma$ compared to SimPO.*

*Table 1.* AlpacaEval 2 (Li et al., 2023) and Arena-Hard (Li* et al., 2024) results under the four settings. LC and WR denote length-controlled and raw win rate, respectively. For AlpacaEval 2, we use the `alpaca_eval_gpt4_turbo_fn` LLM annotator. SimPO (Meng et al., 2024) was retrained; other baselines utilized checkpoints provided by Meng et al. (2024) (see Section 6.3 for full details). We report the standard deviation in Appendix E.

| Method | Mistral-Base (7B) | | | Mistral-Instruct (7B) | | |
| --- | --- | --- | --- | --- | --- | --- |
| | AlpacaEval 2 | | Arena-Hard | AlpacaEval 2 | | Arena-Hard |
| | LC (%) | WR (%) | WR (%) | LC (%) | WR (%) | WR (%) |
| SFT | 4.84 | 2.92 | 1.3 | 24.4 | 18.8 | 12.6 |
| RRHF (Yuan et al., 2023) | 12.7 | 10.2 | 5.8 | 36.0 | 33.0 | 18.1 |
| SLiC-HF (Zhao et al., 2023) | 14.5 | 11.3 | 7.3 | 33.6 | 32.1 | 18.9 |
| DPO (Rafailov et al., 2024b) | 18.1 | 16.0 | 10.4 | 36.1 | 31.9 | 16.3 |
| IPO (Azar et al., 2024) | 13.2 | 10.6 | 7.5 | 28.4 | 26.8 | 16.2 |
| CPO (Xu et al., 2024) | 12.8 | 11.4 | 6.9 | 29.8 | 38.0 | **22.6** |
| KTO (Ethayarajh et al., 2024) | 12.8 | 8.26 | 5.6 | 32.9 | 29.8 | 17.9 |
| ORPO (Hong et al., 2024) | 18.7 | 13.8 | 7.0 | 32.0 | 30.7 | 20.8 |
| R-DPO (Park et al., 2024) | 21.2 | 15.6 | 8.0 | 32.2 | 27.6 | 16.1 |
| SimPO (Meng et al., 2024) | 27.1 | 23.4 | 13.8 | 37.0 | 37.3 | 21.0 |
| **ConfPO (ours)** | **28.9** | **26.7** | **16.9** | **39.1** | **38.4** | 22.4 |

| Method | Llama-3-Base (8B) | | | Llama-3-Instruct (8B) | | |
| --- | --- | --- | --- | --- | --- | --- |
| | AlpacaEval 2 | | Arena-Hard | AlpacaEval 2 | | Arena-Hard |
| | LC (%) | WR (%) | WR (%) | LC (%) | WR (%) | WR (%) |
| SFT | 6.86 | 3.79 | 3.3 | 33.1 | 32.2 | 22.3 |
| RRHF (Yuan et al., 2023) | 13.8 | 9.69 | 6.3 | 35.4 | 32.9 | 26.5 |
| SLiC-HF (Zhao et al., 2023) | 21.4 | 24.3 | 6.0 | 35.7 | 35.6 | 26.2 |
| DPO (Rafailov et al., 2024b) | 25.4 | 23.9 | 15.9 | 43.8 | 42.3 | 32.6 |
| IPO (Azar et al., 2024) | 22.5 | 25.5 | 17.8 | 42.9 | 42.4 | 30.5 |
| CPO (Xu et al., 2024) | 21.9 | 23.0 | 5.8 | 35.7 | 38.7 | 28.8 |
| KTO (Ethayarajh et al., 2024) | 24.7 | 22.1 | 12.5 | 41.0 | 39.7 | 26.4 |
| ORPO (Hong et al., 2024) | 17.6 | 14.2 | 10.8 | 36.7 | 34.7 | 25.8 |
| R-DPO (Park et al., 2024) | 21.9 | 25.0 | 17.2 | 46.2 | 43.9 | **33.1** |
| SimPO (Meng et al., 2024) | 27.0 | 25.7 | 21.1 | 48.8 | 44.4 | 32.6 |
| **ConfPO (ours)** | **28.3** | **32.7** | **23.8** | **49.1** | **45.0** | 32.8 |

# 7. Experimental Results

## 7.1. Main Results and Ablations

**Main results.** Table 1 shows the main results of our experiments. As iterated in Meng et al. (2024), most DPO variants confer minimal performance benefit over standard DPO, whereas SimPO delivers substantial performance gains across all tested models. We introduce ConfPO, a selective-token variant of SimPO that leverages the model's confidence signals for preference learning, which further enhances performance over SimPO on every evaluated model, highlighting the effectiveness of confidence-driven token selection in optimizing preference alignment.

**Selective-token preference learning mitigates overoptimization.** Although Direct Alignment Algorithms (DAAs) do not rely on a separate proxy reward model, Rafailov et al. (2024a) have shown that they still suffer from *overoptimization* (or *reward hacking*). Overoptimization is often illustrated using a Human Alignment vs. Squared KL Divergence plot, where increasing the KL budget beyond a

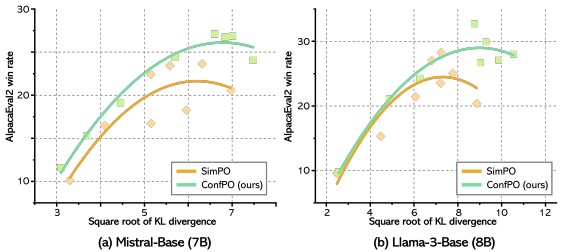

*Figure 7.* **Results on Overoptimization.** Each point in the plots represents a model trained with a different hyperparameter. For both models—(a) Mistral-Base (7B) and (b) Llama-3-Base (8B)—the curve shows that ConfPO places higher than the SimPO, suggesting that it mitigates overoptimization. This improvement is attributed to ConfPO's more efficient use of the KL budget, allowing for better alignment without excessive updates on non-critical tokens.

certain point leads to a decline in model performance (i.e., that is it shows a humped-shaped pattern) (Gao et al., 2023; Rafailov et al., 2024a). Figure 7 presents the overoptimization plots for (a) Mistral-Base (7B) and (b) Llama-3-Base

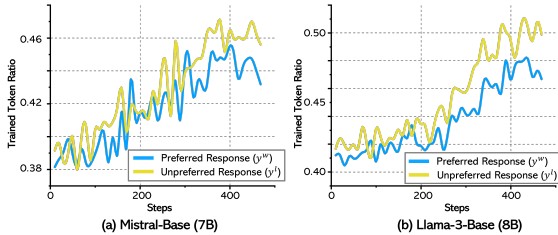

*Figure 8.* **Number of Tokens Selected During Training.** For both models—(a) Mistral-Base (7B) and (b) Llama-3-Base (8B)—the number of tokens selected by our confidence-guided criteria (Eq. 13) gradually increases over the course of training. Notably, ConfPO optimizes fewer than half of the total tokens for preference learning, yet achieves superior performance compared to the baseline, which utilizes all tokens (see Table 1).

(8B). The results indicate that ConfPO effectively mitigates the overoptimization issue observed in SimPO by utilizing the KL divergence budget more efficiently. This could be attributed to the token-selection: since we only perform preference learning with a fraction of tokens that are important for the learning, it prevents meaningless divergence, and thus mitigates the overoptimization.

**Number of tokens selected.** Figure 8 illustrates the proportion of tokens selected throughout training. At the start, approximately **40%** of tokens are selected for both Mistral-Base (7B) and Llama-3-Base (8B). As training progresses, the selection ratio gradually increases, reaching **45%** and **49%**, respectively, by the end. Notably, despite using fewer than half of the total tokens for preference learning, **ConfPO** outperforms SimPO, which utilizes all tokens. This result further underscores that not all tokens contribute equally to preference learning and that selective token optimization can enhance the alignment process.

*Table 2.* AlpacaEval 2 (Li et al., 2023) results for random token selection (ConfPO-rand).

| Method | Mistral-Base (7B) | | Llama-3-Base (8B) | |
|---|---|---|---|---|
| | AlpacaEval 2 | | AlpacaEval 2 | |
| | LC (%) | WR (%) | LC (%) | WR (%) |
| SFT | 4.84 | 2.92 | 6.86 | 3.79 |
| SimPO (Meng et al., 2024) | 27.1 | 23.4 | 27.0 | 25.7 |
| **ConfPO-rand** | 23.3 | 21.4 | 24.0 | 22.8 |
| **ConfPO (ours)** | **28.9** | **26.7** | **28.3** | **32.7** |

**Random token selection does not improve preference learning.** To further assess the effectiveness of our selection strategy, we conduct a control experiment using random token selection while maintaining the same selection ratio as our method. As shown in Table 2, applying preference learning to randomly selected tokens (**ConfPO-rand**) results in worse performance than the baseline (SimPO), which uti-

*Table 3.* AlpacaEval 2 (Li et al., 2023) results comparing DPO and ConfPO applied to DPO (ConfPO$_{\text{DPO}}$ (ours)).

| Method | Mistral-Base (7B) | | Llama-3-Base (8B) | |
|---|---|---|---|---|
| | AlpacaEval 2 | | AlpacaEval 2 | |
| | LC (%) | WR (%) | LC (%) | WR (%) |
| SFT | 8.4 | 6.2 | 1.3 | 17.1 |
| DPO | 18.1 | 16.0 | 25.4 | 23.9 |
| **ConfPO$_{\text{DPO}}$ (ours)** | **23.3** | **19.8** | **29.4** | **27.5** |

lizes all tokens. This outcome highlights the importance of a principled selection strategy, as *simply reducing the number of tokens without informed selection does not yield performance gains.*

### 7.2. Applicability on DPO

Throughout this paper, we have used SimPO (Meng et al., 2024) as our primary baseline. However, the observations presented in Section 4 hold equally for DPO (Rafailov et al., 2024b) (Eq. 4). Supporting plots for DPO can be found in Appendix C. In this section, we extend our **Confidence-Guided Token Selection (ConfPO)** approach to DPO, modifying its objective as follows:

$$\mathcal{L}_{\text{ConfPO}_{\text{DPO}}}(\pi_\theta; \pi_{\text{ref}}) =$$

$$-\mathbb{E}_{(x,y^w,y^l)\sim\mathcal{D}}\Big[\log\sigma\big(\beta\sum_{i=1}^{|y_s^w|} s(y_i^w)\log\frac{\pi_\theta(y_i^w|x,y_{<i}^w)}{\pi_{\text{ref}}(y_i^w|x,y_{<i}^w)}$$

$$-\beta\sum_{i=1}^{|y_s^l|} s(y_i^l)\log\frac{\pi_\theta(y_i^l|x,y_{<i}^l)}{\pi_{\text{ref}}(y_i^l|x,y_{<i}^l)}\big)\Big], \quad (15)$$

where $s(y_i^w) = \mathbf{1}(\pi_\theta(y_i^w|x,y_{<i}^w) \le \tau^w)$, $|y_s^w| = |\{y_i^w|s(y_i^w) = 1\}|$, $s(y_i^l) = \mathbf{1}(\pi_\theta(y_i^l|x,y_{<i}^l) \le \tau^l)$, and $|y_s^l| = |\{y_i^l|s(y_i^l) = 1\}|$. As in Eq. 14, we define the thresholds $\tau^w$ and $\tau^l$ as the average token probabilities.

Table 3 shows that **ConfPO$_{\text{DPO}}$** achieves better human alignment than DPO on the AlpacaEval 2 (Li et al., 2023) benchmark. This demonstrates that our confidence-guided token selection method is not limited to SimPO but can also enhance other Direct Alignment Algorithms (DAAs).

## 8. Conclusion

We introduced **Confidence-Guided Token Selection (ConfPO)**, an approach to preference learning that selectively updates low-confidence tokens, which we identified as crucial for human alignment. Our method improves the performance of state-of-the-art DAAs, including SimPO and DPO, while maintaining the same computational cost. By efficiently utilizing the KL budget and updating fewer than 50% of tokens, ConfPO mitigates overoptimization and enhances alignment without additional overhead.

## Acknowledgments

This work was supported by Artificial intelligence industrial convergence cluster development project funded by the Ministry of Science and ICT(MSIT, Korea)&Gwangju Metropolitan City, Institute for Information & communications Technology Planning & Evaluation (IITP) grant funded by the Korea government(MSIT) (No.RS-2021-II211381, Development of Causal AI through Video Understanding and Reinforcement Learning, and Its Applications to Real Environments), and Institute of Information & communications Technology Planning & Evaluation (IITP) grant funded by the Korea government(MSIT) (No.RS-2022-II220184, Development and Study of AI Technologies to Inexpensively Conform to Evolving Policy on Ethics).

## Impact Statement

This paper does not violate the use of others' work without reference. Furthermore, the paper does not involve introducing new datasets and the experiments conducted do not utilize demographic or identity characteristics. This paper presents work whose goal is to advance the field of Machine Learning. There are many potential societal consequences of our work, none of which we feel must be specifically highlighted here.

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

## A. Limitations

While ConfPO achieves strong empirical results and is grounded in an intuitive rationale, a deeper theoretical analysis is needed to fully elucidate the mechanisms behind its effectiveness. In particular, further investigation is required to formally characterize its impact on human alignment and overoptimization mitigation. Additionally, while we have shown that ConfPO efficiently selects preference-critical tokens for SimPO (Meng et al., 2024) and DPO (Rafailov et al., 2024b), its generalization across different DAAs and model scales remains an open question, warranting further exploration.

*Table 4.* AlpacaEval 2 (Li et al., 2023) results for different choices of threshold $\tau$.

| Method | Mistral-Base (7B) | | Llama-3-Base (8B) | |
|---|---|---|---|---|
| | AlpacaEval 2 | | AlpacaEval 2 | |
| | LC (%) | WR (%) | LC (%) | WR (%) |
| SFT | 4.84 | 2.92 | 6.86 | 3.79 |
| SimPO (Meng et al., 2024) | 27.1 | 23.4 | 27.0 | 25.7 |
| **ConfPO-fixed(0.6)** | 28.2 | 26.2 | 27.6 | 27.5 |
| **ConfPO-geometric** | 27.2 | 26.3 | 28.1 | 28.8 |
| **ConfPO-arithmetic** | **28.9** | **26.7** | **28.3** | **32.7** |

## B. Threshold Selection

In this section, we examine different thresholding strategies for ConfPO (Eq. 13) by evaluating various choices for $\tau$. Specifically, we consider: (1) a fixed threshold ($\tau = 0.6$), (2) the geometric average, and (3) the arithmetic average (which we use in Eq. 14). The geometric average is determined as follows:

$$\tau^w = \left( \prod_{i=1}^{|y^w|} \pi_\theta(y_i^w \mid x, y_{<i}^w) \right)^{\frac{1}{|y^w|}}, \quad \tau^l = \left( \prod_{i=1}^{|y^l|} \pi_\theta(y_i^l \mid x, y_{<i}^l) \right)^{\frac{1}{|y^l|}}. \quad (16)$$

As shown in Table 4, all thresholding strategies outperform standard SimPO, with the arithmetic average yielding the best performance. Given this result, we adopt the arithmetic average for $\tau$ in all experiments.

## C. Observations on DPO

In Figure 9-11, we present observational plots for DPO (Rafailov et al., 2024b) under the same experimental setup as Section 4. The results exhibit trends consistent with those observed in SimPO, further validating our findings. The performance of ConfPO applied to DPO is reported in Table 3 in the main text.

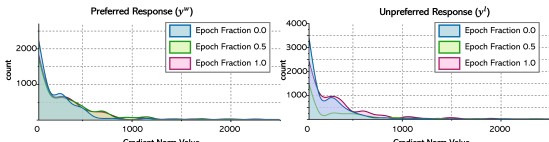

*Figure 9.* **Observation 1 for DPO.**

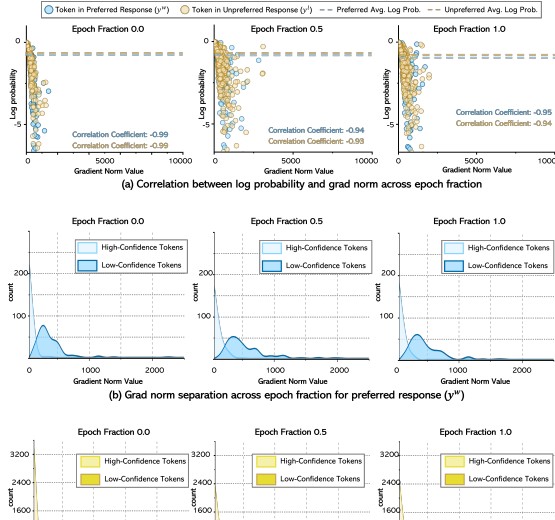

*Figure 10.* **Observation 2 for DPO.**

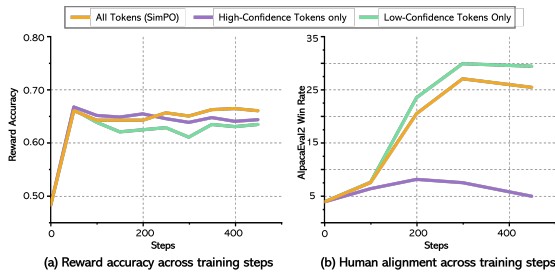

*Figure 11.* **Observation 3 for DPO.**

## D. Implementation Details

Following (Meng et al., 2024), we use a batch size of 128 and train for a single epoch for all preference optimization training. The maximum sequence length is set to 2048, with a maximum prompt length of 1800. We employ a cosine learning rate scheduler with 10% warmup, using AdamW optimizer (Loshchilov & Hutter, 2019). Our implementation is based on TRL (von Werra et al., 2020) and DeepSpeed ZeRO-2, with gradient checkpointing enabled on 4 NVIDIA A100 80GB PCIe GPUs.

For each model, we determine the optimal hyperparameter settings, which are summarized in Table 6. We search the $\beta$ in the range of [1.0, 1.5, 2.0] and $\gamma$ in the range of [0.5, 0.8, 1.2, 1.6, 2.0, 2.5].

*Table 5.* Various preference optimization objectives given preference data $\mathcal{D} = (x, y_w, y_l)$, where $x$ is an input, and $y_w$ and $y_l$ are the winning and losing responses.

| Method | Objective |
|---|---|
| RRHF (Yuan et al., 2023) | $\max\left(0, -\frac{1}{|y_w|}\log\pi_\theta(y_w\mid x) + \frac{1}{|y_l|}\log\pi_\theta(y_l\mid x)\right) - \lambda\log\pi_\theta(y_w\mid x)$ |
| SLiC-HF (Zhao et al., 2023) | $\max\left(0, \delta - \log\pi_\theta(y_w\mid x) + \log\pi_\theta(y_l\mid x)\right) - \lambda\log\pi_\theta(y_w\mid x)$ |
| DPO (Rafailov et al., 2024b) | $-\log\sigma\left(\beta\log\frac{\pi_\theta(y_w\mid x)}{\pi_{\text{ref}}(y_w\mid x)} - \beta\log\frac{\pi_\theta(y_l\mid x)}{\pi_{\text{ref}}(y_l\mid x)}\right)$ |
| IPO (Azar et al., 2024) | $\left(\log\frac{\pi_\theta(y_w\mid x)}{\pi_{\text{ref}}(y_w\mid x)} - \log\frac{\pi_\theta(y_l\mid x)}{\pi_{\text{ref}}(y_l\mid x)} - \frac{1}{2\tau}\right)^2$ |
| CPO (Xu et al., 2024) | $-\log\sigma\left(\beta\log\pi_\theta(y_w\mid x) - \beta\log\pi_\theta(y_l\mid x)\right) - \lambda\log\pi_\theta(y_w\mid x)$ |
| KTO (Ethayarajh et al., 2024) | $-\lambda_w\sigma\left(\beta\log\frac{\pi_\theta(y_w\mid x)}{\pi_{\text{ref}}(y_w\mid x)} - z_{\text{ref}}\right) + \lambda_l\sigma\left(z_{\text{ref}} - \beta\log\frac{\pi_\theta(y_l\mid x)}{\pi_{\text{ref}}(y_l\mid x)}\right),$ 
 where $z_{\text{ref}} = \mathbb{E}_{(x,y)\sim\mathcal{D}}\left[\beta\text{KL}\left(\pi_\theta(y\mid x)\|\pi_{\text{ref}}(y\mid x)\right)\right]$ |
| ORPO (Hong et al., 2024) | $-\log p_\theta(y_w\mid x) - \lambda\log\sigma\left(\log\frac{p_\theta(y_w\mid x)}{1-p_\theta(y_w\mid x)} - \log\frac{p_\theta(y_l\mid x)}{1-p_\theta(y_l\mid x)}\right),$ 
 where $p_\theta(y\mid x) = \exp\left(\frac{1}{|y|}\log\pi_\theta(y\mid x)\right)$ |
| R-DPO (Park et al., 2024) | $-\log\sigma\left(\beta\log\frac{\pi_\theta(y_w\mid x)}{\pi_{\text{ref}}(y_w\mid x)} - \beta\log\frac{\pi_\theta(y_l\mid x)}{\pi_{\text{ref}}(y_l\mid x)} + (\alpha|y_w| - \alpha|y_l|)\right)$ |
| SimPO (Meng et al., 2024) | $-\log\sigma\left(\frac{\beta}{|y_w|}\log\pi_\theta(y_w\mid x) - \frac{\beta}{|y_l|}\log\pi_\theta(y_l\mid x) - \gamma\right)$ |
| **ConfPO (ours)** | $-\log\sigma\left(\frac{\beta}{|y_w|}\sum_{i=1}^{|y^w|}s(y_i^w)\log\pi_\theta(y_i^w\mid x, y_{<i}^w) - \frac{\beta}{|y_l|}\sum_{i=1}^{|y^l|}s(y_i^l)\log\pi_\theta(y_i^l\mid x, y_{<i}^l) - \gamma\right)$ 
 where $s(y_i^w) = \mathbf{1}(\pi_\theta(y_i^w\mid x, y_{<i}^w) \leq \tau^w),\quad |y_s^w| = |\{y_i^w\mid s(y_i^w)=1\}|,$ 
 $s(y_i^l) = \mathbf{1}(\pi_\theta(y_i^l\mid x, y_{<i}^l) \leq \tau^l),\quad |y_s^l| = |\{y_i^l\mid s(y_i^l)=1\}|.$ 
 $\tau^w = \frac{1}{|y^w|}\sum_{i=1}^{|y^w|}\pi_\theta(y_i^w\mid x, y_{<i}^w),\quad \tau^l = \frac{1}{|y^l|}\sum_{i=1}^{|y^l|}\pi_\theta(y_i^l\mid x, y_{<i}^l).$ |

To ensure a fair comparison with existing baselines, we follow the decoding hyperparameters from Meng et al. (2024) for evaluation. Specifically, for AlpacaEval 2, we use sampling-based decoding with temperatures of 0.7, 0.5, and 0.9 for Mistral-Base, Mistral-Instruct, and both LLaMA-3 models, respectively. For Arena-Hard, we apply greedy decoding across all settings.

*Table 6.* The hyperparameter values in ConfPO.

| Setting | $\beta$ | $\gamma$ | Learning rate |
|---|---|---|---|
| **Mistral-Base** | 1.5 | 1.6 | $3e\text{-}7$ |
| **Mistral-Instruct** | 2.0 | 0.5 | $5e\text{-}7$ |
| **Llama-3-Base** | 1.0 | 2.0 | $6e\text{-}7$ |
| **Llama-3-Instruct** | 1.0 | 2.5 | $1e\text{-}6$ |

# E. Standard Deviation of AlpacaEval 2 and Arena-Hardv0.1

Table 7 reports the standard deviation for AlpacaEval 2 and the 95% confidence interval for Arena-Hard.

# F. Correlations between tokens' confidence and their position in the sequences.

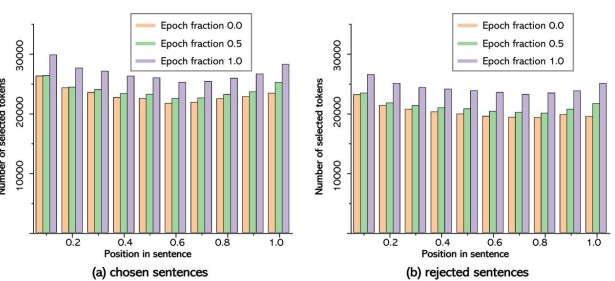

*Figure 12.* Histogram showing the distribution of selected tokens with respect to the position in the sentence.

To check for potential positional biases, we analyzed where low-confidence tokens are selected within a sequence. As shown in Figure 12, the distribution of these tokens is approximately uniform across all normalized positions in both chosen and rejected responses. This pattern holds throughout the training process.

*Table 7.* Detailed statistics for AlpacaEval 2 and Arena-Hard. LC refers to the length-controlled win rate, WR denotes the raw win rate, and STD represents the standard deviation of the win rate. Length indicates the average generation length. For Arena-Hard, we report both the win rate and the 95% confidence interval.

| | AlpacaEval 2 | | | | Arena-Hard | | | |
|---|---|---|---|---|---|---|---|---|
| **Models** | **LC (%)** | **WR (%)** | **STD (%)** | **Length** | **WR** | **95 CI high** | **95 CI low** | **Length** |
| **Mistral-Base** | | | | | | | | |
| **SFT** | 4.84 | 2.92 | 0.6 | 953 | 1.3 | 1.8 | 0.9 | 521 |
| **RRHF** | 12.7 | 10.2 | 1.1 | 1789 | 5.8 | 8.0 | 6.0 | 596 |
| **SLiC-HF** | 14.5 | 11.3 | 1.1 | 1664 | 7.3 | 8.5 | 6.2 | 683 |
| **DPO** | 18.1 | 16.0 | 1.3 | 1732 | 10.4 | 11.7 | 9.4 | 628 |
| **IPO** | 13.2 | 10.6 | 1.1 | 1529 | 7.5 | 8.5 | 6.5 | 674 |
| **CPO** | 12.8 | 11.4 | 1.3 | 1451 | 6.9 | 6.7 | 4.9 | 823 |
| **KTO** | 12.8 | 8.26 | 1.3 | 1208 | 5.6 | 6.6 | 4.7 | 475 |
| **ORPO** | 18.7 | 13.8 | 1.5 | 1582 | 7.0 | 7.9 | 5.9 | 764 |
| **R-DPO** | 21.2 | 15.6 | 1.3 | 1451 | 8.0 | 11.1 | 8.4 | 528 |
| **SimPO** | 27.1 | 23.4 | 1.6 | 1996 | 13.8 | 18.0 | 15.1 | 699 |
| **ConfPO** | 28.9 | 26.7 | 1.6 | 2021 | 16.9 | 18.0 | 15.1 | 699 |
| **Mistral-Instruct** | | | | | | | | |
| **SFT** | 24.4 | 18.8 | 1.3 | 1557 | 12.6 | 14.1 | 11.1 | 486 |
| **RRHF** | 36.0 | 33.0 | 1.5 | 1855 | 18.1 | 19.5 | 16.4 | 517 |
| **SLiC-HF** | 33.6 | 32.1 | 1.5 | 1946 | 18.9 | 20.6 | 17.3 | 578 |
| **DPO** | 36.1 | 31.9 | 1.6 | 1719 | 16.3 | 18.0 | 15.2 | 518 |
| **IPO** | 28.4 | 26.8 | 1.6 | 1949 | 16.2 | 17.9 | 14.4 | 740 |
| **CPO** | 29.8 | 38.0 | 1.6 | 2860 | 22.6 | 25.0 | 20.8 | 812 |
| **KTO** | 32.9 | 29.8 | 1.6 | 2047 | 17.9 | 20.3 | 16.1 | 496 |
| **ORPO** | 32.0 | 30.7 | 1.6 | 2316 | 20.8 | 22.5 | 19.1 | 527 |
| **R-DPO** | 32.2 | 27.6 | 1.6 | 1692 | 16.1 | 18.0 | 14.6 | 495 |
| **SimPO** | 37.0 | 37.3 | 1.7 | 2023 | 21.0 | 22.7 | 18.8 | 539 |
| **ConfPO** | 39.1 | 38.4 | 1.7 | 1981 | 22.7 | 18.0 | 15.1 | 699 |
| **Llama-3-Base** | | | | | | | | |
| **SFT** | 6.86 | 3.79 | 0.7 | 923 | 3.3 | 4.0 | 2.6 | 437 |
| **RRHF** | 13.8 | 9.69 | 1.2 | 1389 | 6.3 | 7.5 | 5.7 | 536 |
| **SLiC-HF** | 21.4 | 24.3 | 1.2 | 1432 | 6.0 | 11.5 | 8.9 | 676 |
| **DPO** | 25.4 | 23.9 | 1.3 | 1732 | 15.9 | 18.1 | 14.1 | 563 |
| **IPO** | 22.5 | 25.5 | 1.5 | 2617 | 17.8 | 19.5 | 16.0 | 608 |
| **CPO** | 21.9 | 23.0 | 1.5 | 2775 | 5.8 | 13.2 | 10.4 | 800 |
| **KTO** | 24.7 | 22.1 | 1.4 | 1782 | 12.5 | 14.2 | 10.9 | 519 |
| **ORPO** | 17.6 | 14.2 | 1.5 | 1599 | 10.8 | 12.3 | 9.6 | 639 |
| **R-DPO** | 21.9 | 25.0 | 1.5 | 3000 | 17.2 | 18.5 | 15.7 | 527 |
| **SimPO** | 27.0 | 25.7 | 1.5 | 1901 | 21.1 | 25.4 | 21.6 | 704 |
| **ConfPO** | 28.3 | 32.7 | 1.6 | 2885 | 23.8 | 18.0 | 15.1 | 699 |
| **Llama-3-Instruct** | | | | | | | | |
| **SFT** | 33.1 | 32.2 | 1.6 | 1949 | 22.3 | 23.9 | 20.3 | 596 |
| **RRHF** | 35.4 | 32.9 | 1.7 | 1893 | 26.5 | 28.4 | 24.6 | 502 |
| **SLiC-HF** | 35.7 | 35.6 | 1.6 | 1728 | 26.2 | 28.4 | 24.4 | 584 |
| **DPO** | 43.8 | 42.3 | 1.7 | 1923 | 32.6 | 34.8 | 30.3 | 528 |
| **IPO** | 42.9 | 42.4 | 1.7 | 2000 | 30.5 | 32.8 | 28.4 | 554 |
| **CPO** | 35.7 | 38.7 | 1.7 | 2148 | 28.8 | 30.6 | 26.6 | 624 |
| **KTO** | 41.0 | 39.7 | 1.7 | 1923 | 26.4 | 28.7 | 24.3 | 536 |
| **ORPO** | 36.7 | 34.7 | 1.6 | 1845 | 25.8 | 27.4 | 23.8 | 535 |
| **R-DPO** | 46.2 | 43.9 | 1.7 | 1904 | 33.1 | 35.3 | 30.9 | 522 |
| **SimPO** | 48.8 | 44.4 | 1.7 | 1795 | 32.6 | 35.9 | 32.0 | 504 |
| **ConfPO** | 49.1 | 45.0 | 1.7 | 1817 | 32.8 | 18.0 | 15.1 | 699 |

