# OpenReview forum: "ConfPO: Exploiting Policy Model Confidence for Critical Token Selection in Preference Optimization"
_ICML.cc/2025/Conference — ICML 2025 poster_

### Official Review · Reviewer_foLE · 2025-03-01

**Overall Recommendation:** 3

**Summary:**

While most existing Direct Alignment Algorithms (DAAs) uniformly adjust token probabilities, this paper questions the assumption that each token contributes equally to preference, and proposes a new method, called ConfPO, which identifies preference-critical tokens based on the training policy's confidence, and zeros out all non-critical tokens in the loss.

Preference-critical tokens are defined as tokens with (conditional) model probabilities below a certain (dynamically defined) threshold. This definition of preference-critical tokens is motivated empirically, as the paper shows a strong negative correlation between model probabilities (i.e., confidence) and gradient norm. Unlike other token-level approaches, a key benefit of this method is that the computation cost remains identical to that of original DAAs. The paper then shows that ConfPO outperforms its DAA analog without critical preference token selection, using the Mistral-7B and Llama-3-8B families of models, and evaluating on AlpacaEval 2 and Arena-Hard v0.1.

**Claims And Evidence:**

The primary claim in the paper is that low-confidence tokens dominate preference learning. This is motivated by the observation that 1) token-level gradient norms follow a long-tailed distribution, and 2) there is a strong negative correlation between gradient norm and confidence. While these observations are supported empirically, there is no theoretical motivation for this simple observation.

The paper also states that ConfPO mitigates overoptimization (reward hacking) by using KL budget more efficiently. While Figure 5 shows a higher win rate for a given KL budget for ConfPO against SimPO at the high end of KL divergence, there is no clear gain at the low end, and this figure alone does not justify the claim about KL budget efficiency (which is nonetheless an intuitive claim).

Also, the claim that "high-confidence tokens can impede preference learning by overshadowing more informative signals from their low-confidence counterparts" provides a nice intuition for why the method works (i.e., a form of denoising), but it was not demonstrated theoretically or empirically that this was the mechanism of ConfPO outperformance.

**Essential References Not Discussed:**

The paper includes the most important prior works in the areas of Direct Alignment Algorithms, including token-level variants.

**Experimental Designs Or Analyses:**

Questions:

1. What is the correlation between tokens' confidence (+ gradient norm) and their position in the sequence? Tokens are generated autoregressively, so are not independent. Are most of the "preference-critical tokens" located towards the beginning, middle, or end of sequences, or are they approximately uniformly distributed? This is a key missing investigation, and it would be nice to see an additional ablation added to Table 2 (beyond ConfPO-rand), where the tokens are selected to match the sequence position distribution of ConfPO.

2. (related to 1) Did you perform any qualitative or semantic analysis of the preference-critical tokens (across both the preferred and dispreferred examples)? Do preference-critical tokens tend to cluster into phrases, or do they tend to be sprinkled across diverse positions in the sequence?

3. ConfPO uses hard weights (thresholding based on token logprobs) in the loss function (equation 11). Did you explore soft weights (e.g. weighting directly proportional to token logprob)? Does performance scale with the "temperature" (so that hard weighting is best)? This would be an insightful ablation.

4. (related to 3) Did you explore a curriculum learning setup, where the weights on preference-critical vs non-preference-critical tokens are adjusted over the course of training? How does this compare to vanilla ConfPO?

**Methods And Evaluation Criteria:**

The experimental setup (including models used and evaluation benchmarks) is pretty straightforward and makes sense. The paper includes reasonable motivation for why ConfPO is built on top of SimPO, and not other DAA variants. The ablation showing that the method also generalizes to DPO provides support for the claim that ConfPO is DAA-agnostic.

The ConfPO method is empirically motivated, based on the observation that tokens with lower probabilities (confidence) tend to have larger gradient norms. But no theoretical justification is provided for this.

**Other Comments Or Suggestions:**

- Section 3.1: Usage of D in equation 2 as the set of prompts differs from the definition stated in the text (preference dataset, with prompts as well as preferred and dispreferred responses). The usage of D in equation 4, on the other hand, aligns with the definition stated in the text.
- Why is the Bradley-Terry model mentioned in Section 3.1? Isn't it more relevant to Section 3.2?
- Figure 4: Which model is trained here? And on which dataset?
- Equation 11 is an exact copy of equation 7, but just going by a different name. Is it necessary to copy, or can r_{ConfPO}(x, y) just be defined as given in equation 7?
- What hyperparameter is being adjusted to plot the points in Figure 5? Especially for Figure 5b, the "Square root of KL divergence" has a notably higher maximum for ConfPO than for SimPO.
- In the "Number of tokens selected" paragraph in Section 7.1, the text states that "at the start, approximately 30% of tokens are selected for Mistral-Base and 40% for Llama-3-Base". But from Figure 6, it looks like the correct number is ~40% in both cases.

**Other Strengths And Weaknesses:**

The ConfPO method is extremely simple and not particularly novel, but the strength of this paper is in the promising results, showing substantial and consistent gains over DAA methods which weight each token equally.

**Questions For Authors:**

No additional questions, besides those mentioned in previous sections. In particular, please see the questions in the "Experimental Designs or Analyses" section.

**Relation To Broader Scientific Literature:**

There has been a recent explosion in literature on Direct Alignment Algorithms, starting with DPO. These algorithms are summarized nicely in Table 5 in the Appendix. The ConfPO algorithm proposed in this paper is primarily explored as a modification to the SimPO algorithm (which modifies DPO by eliminating the need for a reference model and mitigating the length bias). This paper also cites other recent work in token-level preference learning, and differentiates itself by not relying on external models for token-level reward signals.

**Theoretical Claims:**

This was primarily an empirical paper, with no theoretical claims or proofs.

---

> ### Author Rebuttal · Authors · 2025-04-01
>
> Thank you for taking the time to review our paper and for offering detailed feedback. We address them below, following the order in which they appear across the review’s sections.
>
> ---
> ### **Claims and Evidence (CE)**
> **[CE-Q1] Theoretical motivation.**
>
> **[CE-A1]** In our response to reviewer RzSL under question ID [RS-Q1], we provide a detailed theoretical motivation,including mathematical derivations and sensitivity analysis demonstrating why low-confidence tokens ($p_i(\theta)$) yield higher gradient norms ($||\nabla_\theta \log p_i(\theta)||$). Due to space constraints, we kindly refer the reviewer to that section for complete details.
>
>
> **[CE-Q2] KL budget efficiency at lower end.**
>
> **[CE-A2]** We added additional points at lower KL divergence levels in ***Figure E*** to enable a fairer comparison. These additional results clearly show that ConfPO consistently achieves higher alignment scores than SimPO across both low and high KL values.
>
> ***Figure E***: https://anonymous.4open.science/r/ConfPO-B7C6/L/FigureE.md
>
> ---
>
> ### **Experimental Designs or Analyses (EA)**
> **[EA-Q1 & Q2] Correlations between tokens' confidence and their position in the sequences.**
>
> **[EA-A1 & A2]** In ***Figure C*** below, we visualize examples of selected tokens. In general, tokens acting as crucial inflection points within sentences (e.g., tokens initiating lists or new phrases) are selected, while tokens that are merely continuations of preceding tokens are not selected (e.g., in the word "Slack," "Sl" is selected, whereas "ack" is not). This shows that our confidence-based selection identifies important tokens that shape the overall sentence, leading to more efficient KL-budget usage and reduced overoptimization (Figure 5 in the main text).
>
> Since inflection points of sentences are not really located towards certain position in a sentence, we see they are approximately uniformly distributed in the sentence as in ***Figure D***.
>
> ***Figure C***: https://anonymous.4open.science/r/ConfPO-B7C6/L/FigureC.md
>
> ***Figure D***: https://anonymous.4open.science/r/ConfPO-B7C6/L/FigureD.md
>
> **[EA-Q3 & 4] Exploration of soft weighting ConfPO and curriculum learning.**
>
> **[EA-A3 & 4]** Thank you for suggesting this valuable experiment. We explored a soft-weighting version of ConfPO,$$\frac{\sum_{i=1}^{|y|} w(y_i)\log\pi_\theta(y_i|x,y_{<i})}{\sum_{i=1}^{|y|} w(y_i)},$$ where $w(y_i) = (1 - p_i(\theta))^T$. In ***Figure F*** below, we tested soft weighting with temperatures $T=1$ and $5$. When $T=0$, the equation reduces to standard SimPO; as $T$ increases, it approaches our original (hard-weighted) ConfPO. Our results demonstrate that alignment performance consistently improves as $T$ increases, with the original ConfPO achieving the highest performance.
>
> This validates our key hypothesis: high-confidence tokens can impede preference learning by overshadowing more informative signals. This is consistent with Figure 4-(b) of our main paper, where training exclusively on low-confidence tokens improved alignment scores relative to training on all tokens, while training solely on high-confidence tokens resulted in performance worse than even the SFT baseline.
>
> We also experimented with curriculum setups adjusting T from 0 to 10 and 10 to 0 during training. Results showed no clear advantage over standard ConfPO, which can also be found in ***Figure F*** below.
>
> ***Figure F***: https://anonymous.4open.science/r/ConfPO-B7C6/L/FigureF.md
>
> ---
> ### **Other Comments or Suggestions**
> We address each point below:
> - **Definition of \( D \) (Section 3.1):**
> Thank you for noting this inconsistency. We will clearly distinguish the dataset notation used for prompts alone versus prompt-response pairs in the revised manuscript.
> - **Bradley-Terry Model:**
> We included the Bradley–Terry (BT) model in Section 3.1 because it serves as the foundational probabilistic model for preference-based reward modeling, commonly used in RLHF. While traditional RLHF explicitly learns a reward model using BT, DPO (introduced in Section 3.2) leverages this same BT framework implicitly by reparameterizing the reward with the joint log probabilities.
> - **Figure 4 (model and dataset):**
> For Figure2~4, we use the Llama-3-Base (8B) with the UltraFeedback dataset. We will clarify this point in the manuscript.
> - **Equation Duplication:**
> We will simplify the presentation by directly defining as given in Equation 7 with added token selection criteria of ConfPO.
> - **Hyperparameter for Figure 5:**
> In Figure 5, we varied the $\beta$ hyperparameter (with $\gamma$ fixed) to achieve different levels of KL divergence. Also we have added more points in the following figure with fair KL range (Figure E: https://anonymous.4open.science/r/ConfPO-B7C6/L/FigureE.md).
> - **Number of tokens selected:**
> We apologize for the typo. The correct initial selection rate is indeed approximately 40% for both Mistral-Base and Llama-3-Base. We will correct this in the main text.

---

### Official Review · Reviewer_kWCJ · 2025-03-10

**Overall Recommendation:** 3

**Summary:**

This paper proposes a token selection strategy for direct alignment algorithms. They observe a high correlation between the gradient norm and the confidence of a token and use confidence as a metric to automatically select the token, which can possibly prioritize the tokens with high gradient norm. They show empirically that incorporating selection strategy can enhance simpo and DPO algorithms.

**Claims And Evidence:**

Yes. Their claim is supported by clear evidence.

**Essential References Not Discussed:**

None

**Experimental Designs Or Analyses:**

One major weakness is that the evaluation primarily focuses on AlpacaEval Arena-Hard, relying on large language models as judges for human preferences. To better substantiate the effectiveness of the approach, it would be valuable to evaluate its performance on downstream tasks such as reasoning and factual question answering.

**Methods And Evaluation Criteria:**

One weakness is the lack of sufficient baselines to demonstrate the necessity of incorporating dynamic selection. It would be beneficial to include baselines, such as static selection before training.

**Other Comments Or Suggestions:**

None

**Other Strengths And Weaknesses:**

None

**Questions For Authors:**

This paper contributes to the ongoing research on direct alignment algorithms from a feature importance perspective.

1. Since the proposed token selection can be non-continuous, how might this discontinuity impact the coherence or the performance of downstream tasks?

2. Could the authors provide a simple baseline that uses a fixed reference policy to select important tokens? This would help demonstrate the necessity of dynamic selection.

3. A key implicit assumption is that low-confidence tokens contain more useful information and lead to more effective updates. However, if the data contains noise, could this selection mechanism amplify it, potentially resulting in degenerated performance?

4. This selection mechanism appears to be applicable in both the base and instruct settings, where the former involves off-policy data and the latter on-policy data to some extent. I am curious whether this selection method is also applicable to general RLHF algorithms, such as PPO, where all samples are highly on-policy. In this scenario, tokens are more likely to have high confidence due to the on-policy nature of the data generation.

**Relation To Broader Scientific Literature:**

This paper contributes to the ongoing research on direct alignment algorithms from a feature importance perspective.

**Theoretical Claims:**

This work focuses on empirical analysis.

---

> ### Author Rebuttal · Authors · 2025-04-01
>
> Thank you for taking the time to review our paper and for offering detailed feedback. We address them below, following the order in which they appear across the review’s sections.
>
> ---
> ### **Methods and Evaluation Criteria (ME)**
> **[ME-Q1] Static token selection baseline.**
>
> **[ME-A1]** We address this question under *[QA-Q2]* below, where we provide further details on the static token selection baseline.
>
> ---
> ### **Theoretical Claims (TC)**
> **[TC-Q1] Theoretical Motivation**
>
> **[TC-A1]** In our response to reviewer RzSL under question *[SW-Q1]*, we provide a detailed theoretical motivation, including mathematical derivations and sensitivity analysis demonstrating why low-confidence tokens ($p_i(\theta)$) yield higher gradient norms ($||\nabla_\theta \log p_i(\theta)||$). Due to space constraints, we kindly refer the reviewer to that section for complete details.
>
> ---
> ### **Experimental Designs or Anlayses (EA)**
> **[EA-Q1] Evaluation on downstream tasks**
>
> **[EA-A1]** We have evaluated our trained models on downstream tasks involving math, truthfulness, reasoning, and coding. We find that ConfPO achieves the highest average performance across all baselines. Due to space constraints, please refer to our response to reviewer M4QG under question *[ME-Q1]* or at the following link: https://anonymous.4open.science/r/ConfPO-B7C6/L/TableB.md
>
> ---
> ### **Supplementary Material (SM)**
> **[SM-Q1] Typo in Table 7.**
>
> **[SM-A1]** Thank you for pointing out the typo. We have corrected Table 7, which is available at the following link: https://anonymous.4open.science/r/ConfPO-B7C6/L/TableF.md
>
> ---
> ### **Questions for Authors (QA)**
> **[QA-Q1] Concern about non-continuous update on coherency and performance of downstream task.**
>
> **[QA-A1]** Because each token is generated under a conditional distribution that considers all previously generated tokens, selectively updating some tokens while skipping others does not fragment the model’s coherence. Despite directing the preference objective toward challenging (low-confidence) tokens, the model still maintains a valid joint distribution $p(y)$ at inference time, ensuring coherent token-by-token generation. Empirically, we do not observe drops in fluency or logical consistency on bechmarks like AlpacaEval 2, Arena-Hard, and downstream tasks shown in question *[EA-Q1]* above. A small-scale human assessment of responses also further confirms that selectively omitting updates on high-confidence tokens does not undermine text coherency.
>
> **[QA-Q2] Static token selection baseline.**
>
> **[QA-A2]** We conducted an additional baseline experiment, selecting tokens based on confidence values from a fixed reference policy. Our results (see ***Table G***) indicate that approximately 80% of tokens selected dynamically by the evolving policy model overlapped with tokens chosen by the fixed reference policy; however, the remaining 20% differed, reflecting critical tokens that changed dynamically as training progressed. While token selection via the fixed reference policy slightly improved performance over SimPO, it was noticeably inferior compared to our dynamic ConfPO method. This clearly highlights that token importance evolves as the policy model updates, underscoring the necessity of our dynamic selection strategy for achieving optimal performance.
>
> ***Table G***: https://anonymous.4open.science/r/ConfPO-B7C6/L/TableG.md
>
> **[QA-Q3] Does noisy data result in degenerated performance.**
>
> **[QA-A3]** Thank you for raising this important concern. We tested ConfPO with two types of noise:
>
> 1. **Flipped Preferences:** We randomly flipped 20% of the preference labels. Even in this noisy setting, ConfPO outperformed SimPO, confirming our low-confidence token selection does not amplify label noise.
>
> 2. **Word-Level Noise (EDA):** We also introduced word-level noise via data augmentation methods—synonym replacement, random insertion, random swap, and random deletion. Although this aggressive augmentation substantially degraded overall performance, ConfPO still outperformed SimPO.
>
> The detailed results are available at the following link:
>
> ***Table H***: https://anonymous.4open.science/r/ConfPO-B7C6/L/TableH.md
>
> We note that word-level noise is uncommon in recent preference-learning scenarios, which typically use on-policy data collection from advanced LLM-generated text that produces coherent sentences. Thus, randomly flipped preferences represent a more realistic noise scenario.
>
> **[QA-Q4] Applicability to on-policy RLHF, such as PPO.**
>
> **[QA-A4]** Our token-selection mechanism indeed applies effectively to both off-policy and partially on-policy settings, as shown. Even with highly on-policy data, our dynamic threshold maintains stable token selection ratios. Extending ConfPO explicitly to PPO-based RLHF, however, involves reward-based signals rather than implicit log-probabilities used by DAAs, and thus lies beyond our current scope but remains promising future work.

---

> > ### Comment · Reviewer_kWCJ · 2025-04-02
> >
> > Thank you for the responses. They address my main concerns, and I will maintain my current score.

---

### Official Review · Reviewer_RzSL · 2025-03-14

**Overall Recommendation:** 2

**Summary:**

This paper proposes to only include low-confidence tokens during preference alignment.

**Claims And Evidence:**

Yes.

**Essential References Not Discussed:**

No.

**Experimental Designs Or Analyses:**

Yes.

1. Llama-3 and Mistral seem to be outdated. It would be better using more advanced models like Llama-3.1, Deepseek and Qwen.

2. It will be better to have diverse tasks like math and coding.

**Methods And Evaluation Criteria:**

Yes.

**Other Comments Or Suggestions:**

No

**Other Strengths And Weaknesses:**

Weaknesses:

1. The relationship between gradient norm and confidence is weak, only supported  by limited empirical experiment observation.

2. It would be beneficial to select specific examples and provide visualizations of the chosen tokens. This could reveal whether the selected tokens capture core information or merely auxiliary words.

3. This actually corresponds to all alignment algorithms. So it would be better to see the combination of this selection strategy with different algorithms.

4. The novelty is limited as it just slightly modifies the loss without much theoretical justification. Also, the performance is similar to existing works.

**Questions For Authors:**

No

**Relation To Broader Scientific Literature:**

This is a slight modification of SimPO without much novelty and contribution.

**Theoretical Claims:**

There is no theory in this work/

---

> ### Author Rebuttal · Authors · 2025-04-01
>
> Thank you for taking the time to review our paper and for offering detailed feedback. We address them below, following the order in which they appear across the review’s sections.
>
> ---
> ### **Experimental Designs or Analyses (EA)**
> **[CE-Q1] Experiment on newer models.**
>
> **[CE-A1]** We focused on the four models used by SimPO, as these models had thoroughly tuned hyperparameters for nine existing DAAs, ensuring a fair comparison.
>
> However, following the reviewer’s suggestion, we provide additional experimental results using Qwen2.5 7B, shown in ***Table E*** below, where we see ConfPO has higher score than SimPO.
>
> ***Table E***: https://anonymous.4open.science/r/ConfPO-B7C6/L/TableE.md
>
> **[CE-Q2] Evaluation on diverse tasks**
>
> **[CE-A2]** We have evaluated our trained models on various downstream tasks, including math and coding, and find that ConfPO achieves the highest average performance than baselines. Due to space constraints, please refer to our response to reviewer M4QG under question [ME-Q1] or at the following link: https://anonymous.4open.science/r/ConfPO-B7C6/L/TableB.md
>
> ---
> ### **Strengths and Weaknesses (SW)**
> **[SW-Q1] Theoretical Motivation.**
>
> **[SW-A1]** For theoretical clarity, we provide a first-principles analysis showing why low-confidence tokens naturally yield higher gradient norms.
>
> Let $\pi_\theta$ be our policy model, parameterized by $\theta$. Given a prompt $x$ and a response $y =\{y_1,\dots,y_n\}$, let $$p_i(\theta)=\pi_\theta(y_i|x,\,y_{<i})$$be the probability of the $i$-th token. Our goal is to show that the gradient norm of the log-probability of the $i$-th token, $||\nabla_\theta\log p_i(\theta)||$, is **negatively correlated** with the confidence of the policy model for that token, $p_i(\theta)$.
>
> **Chain Rule and Gradient Norm**
>
> Using the chain rule, we have:
> $$\nabla_\theta\log p_i(\theta)=\frac{\nabla_\theta p_i(\theta)}{p_i(\theta)}\implies||\nabla_\theta\log p_i(\theta)||=\frac{||\nabla_\theta p_i(\theta)||}{p_i(\theta)}.$$
> Thus, if token confidence $p_i(\theta)$ dominates this ratio, the gradient norm of the log probability $\nabla_\theta\log p_i(\theta)$ decreases as confidence increases, establishing the observed negative correlation.
>
> **Sensitivity Analysis**
>
> To formalize the dominance of $p_i(\theta)$ on $\nabla_\theta \log p_i(\theta)$, consider the ratio $$r(\theta)=\frac{||\nabla_\theta p_i(\theta)||}{p_i(\theta)}=\frac{b(\theta)}{c(\theta)},$$where $b(\theta)=||\nabla_\theta p_i(\theta)||$ and $c(\theta)=p_i(\theta)$. A local sensitivity test examines partial derivatives:$$\frac{\partial r}{\partial b}=\frac{1}{c},\quad\frac{\partial r}{\partial c}=-\frac{b}{c^2}.$$To determine which component more strongly influences the ratio, one can sample many tokens $ i $, each yielding pairs $(b_i, c_i)$. Then, examine the *main effects*:
> $$S_b(i)=\frac{1}{c_i},\quad S_c(i) =-\frac{b_i}{c_i^2}.$$
> Comparing $\mathbb{E}[|S_b|]$ and $ \mathbb{E}[|S_c|] $ offers a practical measure of which component dominates in typical scenarios. If $\mathbb{E}[|S_c|-|S_b|]>0$, it shows that variations in $ p_i(\theta) $ (the token’s confidence) have the stronger effect on $||\nabla_\theta \log p_i(\theta)||$. Using Monte Carlo estimate with 1000 samples, we indeed find that $\mathbb{E}[|S_c|-|S_b|]>0$ (The exact value can be found in https://anonymous.4open.science/r/ConfPO-B7C6/L/TableI.md). Thus, this describes our observed **negative correlation** between the confidence and log-probability gradient norm.
>
> **[SW-Q2] Visualization of selected tokens.**
>
> **[SW-A2]** In ***Figure C*** below, we visualize examples of selected tokens. In general, tokens acting as crucial inflection points within sentences (e.g., tokens initiating new phrases) are selected, while tokens that are merely continuations of preceding tokens are not selected (e.g., in the word "Slack," "Sl" is selected, whereas "ack" is not). This shows that our confidence-based selection identifies important tokens that shape the overall sentence, leading to more efficient KL-budget usage and reduced overoptimization (Figure 5 in the main text).
>
> ***Figure C***: https://anonymous.4open.science/r/ConfPO-B7C6/L/FigureC.md
>
> **[SW-Q3] Extension to other DAAs.**
>
> **[SW-A3]** In our main text, we showed how ConfPO applies to SimPO (Table 1) and DPO (Table 3). Additionally, we successfully extended ConfPO to IPO with enhanced performance (see response to reviewer M4QG, *[RS-Q2]*). Due to the space limit, we kindly direct the reviewer to that section or to the following link: https://anonymous.4open.science/r/ConfPO-B7C6/L/TableD.md
>
> **[SW-Q4] Theoretical Motivation and Performance.**
>
> **[SW-A4]** Please refer to [SW-Q1] for the theoretical motivation. In terms of the performance, our method consistently improves on existing DAAs (SimPO, DPO, IPO) across multiple benchmarks. For instance, on Llama-3-Base(8B) for AlpacaEval2, ConfPO achieves a +7% win rate over SimPO, showing that our modified loss leads to meaningful alignment gains.

---

> > ### Comment · Reviewer_RzSL · 2025-04-02
> >
> > I appreciate the authors’ rebuttal, but I remain unconvinced. The proposed "theory" does not make sense to me. The chain rule cannot demonstrate negative correlation, as the numerator also varies. I fail to see any logic in the "sensitivity analysis" and its connection with negative correlation. In fact, I don’t believe a theoretical explanation is necessary for this claim. The idea itself is quite intuitive—for instance, non-critical tokens like auxiliary words often have high confidence but contribute little to the gradient. Based on this, the authors suggest focusing only on low-confidence tokens, as they are more likely to be critical. However, I doubt the robustness of this approach across all algorithms and datasets. Important information may be overlooked, and certain key details appear to be missing.

---

> > > ### Author Response · Authors · 2025-04-03
> > >
> > > ### Response to Rebuttal Comment by Reviewer RzSL
> > >
> > > We thank the reviewer for the rebuttal comment and would like to clarify the raised concern as follows:
> > >
> > > 1. **“The chain rule alone cannot demonstrate negative correlation.”**
> > >    We do not rely solely on the chain rule to conclude negative correlation. Rather, the chain rule
> > >    $$
> > >    \nabla_\theta \log p_i(\theta)
> > >    = \frac{\nabla_\theta p_i(\theta)}{p_i(\theta)}
> > >    $$
> > >    simply reveals that $||\nabla_\theta \log p_i(\theta)||$ depends on the inverse of to $p_i(\theta)$. The key point is *how strongly* changes in $p_i(\theta)$ versus changes in $||\nabla_\theta p_i(\theta)||$ affect the ratio. This motivates our **sensitivity analysis**, which goes beyond the chain rule to quantify which term dominates in practice.
> > >
> > > ---
> > >
> > > 2. **“Lack of logic in the sensitivity analysis and its connection to correlation.”**
> > >     ***Local sensitivity analysis*** is a widely adopted statistical method that quantifies how variations in individual input variables affect a target output. Specifically, in our context, it would be seeing how $||\nabla_\theta p_i(\theta)||$ and $p_i(\theta)$ affect the ratio  $r(\theta) = \frac{||\nabla_\theta p_i(\theta)||}{p_i(\theta)}$, through their respective partial derivatives.
> > >
> > >    We decompose $r$ into a numerator $b(\theta) = ||\nabla_\theta p_i(\theta)||$ and denominator $c(\theta) = p_i(\theta)$ and compute the partial derivatives: $\frac{\partial r}{\partial b} = \frac{1}{c}, \quad \frac{\partial r}{\partial c} = -\frac{b}{c^2}$. Empirically, we sample many tokens and compare the “main effects” $\tfrac{1}{c_i}$ vs. $-\tfrac{b_i}{c_i^2}$. If the latter term (tied to $c_i = p_i(\theta)$) consistently dominates, it indicates that larger $p_i(\theta)$ values drive the ratio down more than variations in $||\nabla_\theta p_i(\theta)||$ push it up. This yields a net *negative* relationship between $p_i(\theta)$ and $||\nabla_\theta \log p_i(\theta)||$.
> > >
> > > ---
> > >
> > > 3. **“A theoretical explanation seems unnecessary; the idea is intuitive.”**
> > > We agree that the phenomenon is intuitive: As seen in ***Figure C***, highly "confident" tokens often contribute less information, typically serving as straightforward continuations of preceding tokens, whereas low-confidence tokens tend to represent important inflection points in sentences. **However**, we believe that formalizing *why* this occurs helps rigorously validate the intuition and strengthens the theoretical grounding of our method.
> > >
> > >     ***Figure C***: https://anonymous.4open.science/r/ConfPO-B7C6/L/FigureC.md
> > >
> > > ---
> > >
> > > 4. **“Robustness across all algorithms/datasets.”**
> > > In our paper, we explored four different models (Mistral-base 7B, Llama-base 8B, Mistral-instruct 7B, Llama-instruct 8B), along with an additional model (Qwen 2.5 7B) tested during this rebuttal (see question [CE-Q1] or Table E: ***Table E***: https://anonymous.4open.science/r/ConfPO-B7C6/L/TableE.md), and three different datasets: UltraFeedback (for Mistral-base 7B, Llama-base 8B, and Qwen2.5 7B), on-policy data for Mistral-instruct 7B, and on-policy data for Llama-instruct 8B.
> > >
> > >    Furthermore, we demonstrated the robustness of ConfPO by applying it across multiple DAAs—SimPO (Table 1), DPO (Table 3), and IPO (see question [SW-Q3] or Table D: ***Table D***: https://anonymous.4open.science/r/ConfPO-B7C6/L/TableD.md).
> > >
> > > **We would appreciate it if the reviewer could explicitly indicate which aspects of our response regarding the sensitivity analysis or the experimental results remain unclear, so that we can provide further clarification.**

---

### Official Review · Reviewer_M4QG · 2025-03-16

**Overall Recommendation:** 2

**Summary:**

This paper introduces ConfPO, a novel method for enhancing preference learning in large language models (LLMs). The core idea behind ConfPO is to selectively update tokens during training based on their confidence levels, specifically focusing on low-confidence tokens which the authors empirically demonstrate are crucial for effective alignment. The method is motivated by three key observations: that token-level gradients follow a long-tailed distribution, there is a high correlation between the gradient norm and a token's confidence, and that low-confidence tokens dominate the learning process during preference optimization. The authors propose a simple yet effective approach where only tokens with a probability lower than the average probability across the sequence are selected for backpropagation. Specifically, the authors extend their method to DPO, demonstrating its applicability across different DAA frameworks.

Empirically, ConfPO demonstrates superior performance compared to existing DAAs, including SimPO and DPO, on benchmarks such as AlpacaEval 2 and Arena-Hard. The results indicate that ConfPO achieves higher win rates and reduces the risk of reward hacking. The authors conduct experiments using Mistral 7B and Llama-3-8B models, in both base and instruct configurations, and evaluate their method on the AlpacaEval 2 and Arena-Hard benchmarks. They show that ConfPO improves alignment performance and mitigates overoptimization, a common issue in preference learning. The paper also includes an ablation study in the appendix, which explores different thresholding strategies for selecting low-confidence tokens, finding that the average probability threshold performs best.

**Claims And Evidence:**

The paper claims that ConfPO has the same computational cost as standard preference learning, but it lacks quantitative data to support this claim. While the authors mention that their method is robust to hyperparameter choices, this claim should be supported by more extensive experimentation.

**Essential References Not Discussed:**

No.

**Experimental Designs Or Analyses:**

The authors propose ConfPO which only selects tokens with a probability lower than the average probability. Experiments show that ConfPO improves alignment performance and mitigates overoptimization. Howeve, this paper lacks the essential comparison with existing work using similar ideas, such as TDPO [1] and SparsePO [2].

[1] Token-level Direct Preference Optimization
[2] SPARSEPO: CONTROLLING PREFERENCE VIA SPARSE TOKEN MASKS

**Methods And Evaluation Criteria:**

The authors should also consider using more fine-grained evaluation metrics that can capture the nuances of model behavior on these tasks. For example, they could use metrics that measure the quality of reasoning or the correctness of generated code. This would provide a more detailed assessment of ConfPO's performance and help identify areas for further improvement.

**Other Comments Or Suggestions:**

No.

**Other Strengths And Weaknesses:**

No.

**Questions For Authors:**

The paper uses the average probability as a threshold, but it is unclear whether this is the most effective approach in all cases. A more detailed analysis of different thresholding strategies and their impact on performance is needed. Third, how sensitive is ConfPO to variations in other hyperparameters, such as the learning rate and batch size, and how do these hyperparameters interact with each other?

What is the computational cost of ConfPO compared to other alignment techniques, such as DPO and SimPO, in terms of training time, memory usage, and energy consumption?

**Relation To Broader Scientific Literature:**

The paper primarily focuses on the SimPO algorithm, and while it does extend the method to DPO, a more comprehensive comparison with other DAAs would strengthen the paper's claims. A deeper investigation beyond token selection into the theoretical underpinnings of why ConfPO works across various DAAs, beyond empirical observation, is needed.

**Theoretical Claims:**

Yes. I checked the objective function listed in Table 5.

---

> ### Author Rebuttal · Authors · 2025-04-01
>
> Thank you for taking the time to review our paper and for offering detailed feedback. We address them below, following the order in which they appear across the review’s sections.
>
> ---
> ### **Claims and Evidence (CE)**
> **[CE-Q1] Computational cost comparison.**
>
> **[CE-A1]** **Table A1** compares runtime and GPU memory usage of ConfPO with baseline DAAs (DPO, SimPO). As ConfPO requires no extra forward/backward passes or auxiliary models, its computational cost is nearly identical to the baselines.
>
> **Table A1.** Runtime and peak GPU memory usage for each method under identical settings.
> ||Time/Iter(s)|Mem(GB)|
> |-|-|-|
> |DPO|121.8|53GB|
> |ConfPO$_{DPO}$|125.8|51GB|
> |SimPO|150.2|50GB|
> |ConfPO|140.0|49GB|
>
> **[CE-Q2] Robustness to hyperparameter choices.**
>
> **[CE-A2]** We want to clarify that we do not claim our method is “robust” to hyperparameters. Rather, in Figure 5 (where we vary the KL-divergence of resulting models by adjusting the hyperparameters), it indicates our method remains more stable when increasing the KL budget which is a key concern in reward-hacking (overoptimization) scenarios. We will revise the manuscript to avoid any confusion about hyperparameter robustness and emphasize that these experiments were specifically designed to demonstrate reduced overoptimization.
>
> However, with regards to this, we have attached ***Figure A*** below, comparing ConfPO and SimPO across various hyperparameter settings.
>
> ***Figure A***: https://anonymous.4open.science/r/ConfPO-B7C6/L/FigureA.md
>
> ---
> ### **Methods and Evaluation Criteria (ME)**
> **[ME-Q1] Evaluation on downstream tasks**
>
> **[ME-A1]** We have evaluated our trained models on downstream tasks involving math, QA, reasoning, and coding. We find that ConfPO achieves the highest average performance across all baselines. Due to space constraints, we summarize key results below (***Table A2***) and provide the full table (***Table B***) in the following link:
>
> ***Table B***: https://anonymous.4open.science/r/ConfPO-B7C6/L/TableB.md
>
> **Table A2.** Downstream task results on Llama-3-Base (8B).
> ||GSM 8K (Math)|TruthfulQA|ARC (Reasoning)|Humaneval (Coding)|
> |-|-|-|-|-|
> |**Llama-3-Base (8B)**|||||
> |SFT|51.1|45.3|55.6|6.71|
> |DPO|51.2|53.5|56.1|12.5|
> |SimPO|50.8|56.2|56.7|12.8|
> |ConfPO|**52.9**|**59.2**|**58.4**|**17.0**|
>
> ---
> ---
> ### **Experimental Designs or Analyses (EA)**
> **[EA-Q1] Comparison with existing token-level DAAs**
>
> **[EA-A1]** We include comparisons with TDPO and SparsePO in ***Table C*** below. While TDPO outperforms SimPO by leveraging token-level signals, our ConfPO still achieves higher alignment performance. Notably, TDPO requires a reference model to obtain token-level signals, incurring higher memory and compute overhead compared to ConfPO, which can identify critical tokens without a reference model. Meanwhile, SparsePO demonstrates performance comparable to SimPO.
>
> ***Table C***: https://anonymous.4open.science/r/ConfPO-B7C6/L/TableC.md
>
> ---
> ### **Relation to Broader Scientific Literature (RS)**
>
> **[RS-Q1] Theoretical Motivation**
>
> **[RS-A1]** In our response to reviewer RzSL under question *[SW-Q1]*, we provide a detailed theoretical motivation,including mathematical derivations and sensitivity analysis demonstrating why low-confidence tokens ($p_i(\theta)$) yield higher gradient norms ($||\nabla_\theta \log p_i(\theta)||$). Due to space constraints, we kindly refer the reviewer to that section for complete details.
>
> Furthermore, we emphasize that various DAAs utilize the joint log probability as an implicit reward, causing their policy to fundamentally depend on $\nabla_\theta \log p_i(\theta)$. Consequently, our theoretical insights naturally generalize across various DAAs.
>
> **[RS-Q2] Extension to more DAAs**
>
> **[RS-A2]** In our main text, we have shown ConfPO to work on both the DPO and SimPO. We have now also applied ConfPO to IPO in ***Table D*** below, supporting the general applicability of ConfPO on various DAAs.
>
> ***Table D***: https://anonymous.4open.science/r/ConfPO-B7C6/L/TableD.md
>
> ---
> ### **Questions for Authors (QA)**
> **[QA-Q1] Analysis of different thresholding strategies.**
>
> **[QA-A1]** In Appendix B (Table 4), we analyze various thresholding strategies (fixed, geometric, arithmetic averages) and find that the arithmetic average, as used in our main experiments, provides the best performance.
>
> **[QA-Q2] How sensitive is ConfPO to variations in other hyperparameters such as learning rate?**
>
> **[QA-A2]** We have included ***Figure B*** below comparing ConfPO with SimPO across different learning rates (lr). Due to SimPO's design (no explicit KL regularization), it tends to become unstable at higher learning rates. In contrast, ConfPO, by selectively updating only critical tokens, demonstrates increased stability, even at higher lr.
>
> ***Figure B***: https://anonymous.4open.science/r/ConfPO-B7C6/L/FigureB.md
>
> **[QA-Q3] Computational cost comparison.**
>
> **[QA-A3]** We have provided a detailed comparison under **[CE-Q1]** above.

---

### Decision · Program_Chairs · 2025-05-01

**Decision:**

Accept (poster)

**Comment:**

My recommendation is to accept the paper.

The paper proposes filtering out high-confidence tokens when performing gradient updates in direct alignment algorithms such as DPO. The authors show that the ConfPO method outperforms current baselines and appears to be less prone to reward hacking. Several rationales are given for the filtering, and the filtering method is applied to multiple DAAs.

Reviewers raised several questions about the motivation of the method, comparisons to other methods, and applicability to other DAAs. The authors addressed these questions well in the rebuttal. I would encourage the authors to include these disucssions and comparisons in the text of the paper.